# Alternative proteoforms and proteoform-dependent assemblies in humans and plants

Claire D McWhite [ID][1✉], Wisath Sae-Lee [ID][2], Yaning Yuan [ID][3], Anna L Mallam[2],
Nicolas A Gort-Freitas [ID][4], Silvia Ramundo[5], Masayuki Onishi [ID][3] & Edward M Marcotte [ID][2]

## Abstract

**The variability of proteins at the sequence level creates an enormous potential for proteome complexity. Exploring the depths and limits of this complexity is an ongoing goal in biology. Here, we systematically survey human and plant high-throughput bottom-up native proteomics data for protein truncation variants, where substantial regions of the full-length protein are missing from an observed protein product. In humans, *Arabidopsis*, and the green alga *Chlamydomonas*, approximately one percent of observed proteins show a short form, which we can assign by comparison to RNA isoforms as either likely deriving from transcript-directed processes or limited proteolysis. While some detected protein fragments align with known splice forms and protein cleavage events, multiple examples are previously undescribed, such as our observation of fibrocystin proteolysis and nuclear translocation in a green alga. We find that truncations occur almost entirely between structured protein domains, even when short forms are derived from transcript variants. Intriguingly, multiple endogenous protein truncations of phase-separating translational proteins resemble cleaved proteoforms produced by enteroviruses during infection. Some truncated proteins are also observed in both humans and plants, suggesting that they date to the last eukaryotic common ancestor. Finally, we describe novel proteoform-specific protein complexes, where the loss of a domain may accompany complex formation.**

**Keywords** Proteoform; Proteolytic Processing; Protein Evolution; Co-fractionation/Mass Spectrometry (CF/MS); Alternative Splicing
**Subject Category** Proteomics

## Introduction

Proteoforms are protein isoforms encoded by the same gene that vary in either amino acid sequence or post-translational modification (Smith and Kelleher, 2013). One class of proteoforms is truncation variants, where an observed protein product is missing a substantial portion of its genetically encoded sequence. The full extent of protein variation in the cell is unknown (Aebersold et al, 2018), and the number of proteins that exist as both long and short forms is similarly unknown, although many have been discovered historically.

Some short proteoforms are derived from alternate transcription, as in the case of the multifunctional enzyme PUR2, which is alternately transcribed as a complete tri-enzyme, or a short proteoform consisting of only its first enzyme (Henikoff, Sloan, and Kelly, 1983). Other short proteoforms derive from post-translational limited proteolysis, as in the case of the signaling protein NOTCH1, which is proteolyzed upon activation into two short proteoforms, one that translocates to the nucleus and another that remains embedded in the cell membrane (Kopan and Ilagan, 2009). Limited/regulatory proteolysis occurs at a specific position in the protein chain, and is a distinct process from non-specific proteolytic degradation. While transcription-derived protein truncation variants may be predicted from mRNA isoforms (as for PUR2), proteoforms arising from limited post-translational proteolysis cannot easily be predicted from transcriptomes (as for NOTCH1).

Detection of a short proteoform can transform our understanding of a protein's function, as was the case with the discovery of NOTCH1's dual role as a membrane receptor and nuclear transcription factor. However, techniques that have been traditionally used to discover short proteoforms, e.g., size separation by gel electrophoresis and western blots (Löffler and Huber, 1992), typically allow the characterization of only a single protein at a time. Individual protein assays are not always easily scalable to whole proteomes, especially for species where genetic tagging is not feasible. Numerous high-throughput approaches have been developed to detect proteoforms based on "top-down" analysis of small intact proteins using mass spectrometry (e.g., as in (Dai et al, 2019; Ansong et al, 2013; Tran et al, 2011; Shortreed et al, 2016)), however, these methods are roughly limited to proteoforms with a mass of less than ~50 kDa.

With the average mass of human protein ~62 kDa, alternate methods are necessary to identify longer proteoforms at high throughput.

Alternate methods first separate proteins by size, then identify size variants by conventional "bottom-up" mass spectrometry,

[1]Lewis-Sigler Institute for Integrative Genomics, Princeton University, Princeton, NJ 08544, USA. [2]Department of Molecular Biosciences, The University of Texas at Austin, Austin, TX 78712, USA. [3]Department of Biology, Duke University, Durham, NC 27708, USA. [4]Department of Systems Biology, Harvard Medical School, Boston, MA 02115, USA. [5]Gregor Mendel Institute of Molecular Plant Biology, 1030 Wien, Austria. ✉E-mail: cmcwhite@princeton.edu

where proteoforms are identified by peptide coverage. For example, in the PROTOMAP approach, proteins are separated by 1D-SDS-PAGE, then gel lanes are cut into slices for protein (and associated peptide) detection (Dix et al, 2008). Initially used to detect proteolytic processing events based on finding N- or C-terminal short proteoforms, PROTOMAP has been broadly applied, including to the discovery of short splice forms of human aminoacyl tRNA synthetases (Lo et al, 2014). Other notable strategies include the use of Combined Fractional Diagonal Chromatography (COFRADIC) to specifically isolate N- or C-terminal peptides (Gevaert et al, 2003; Staes et al, 2011; Van Damme et al, 2014; Tanco et al, 2017), and the related methods of COrrelation-based functional ProteoForm (COPF) assessment and Peptide Correlation Analysis (PeCorA), in which proteoforms are detected based on the correlation among peptide abundances observed across bottom-up proteomics experiments (Bludau et al, 2021; Dermit et al, 2021).

In our methodology (Fig. 1A), we first fractionate protein lysate into fractions by, i.e., native size exclusion or ion exchange chromatography and identify peptides in each fraction by mass spectrometry (McWhite et al, 2021). The concept of separating proteins/proteoforms by their distinct physical properties is highly analogous to the standard method by which many endogenous truncated proteoforms were discovered, in which proteins were separated electrophoretically and detected by e.g., Western blotting. However, coupling biochemical fractionation with mass spectrometry allows us to record the elution behavior of thousands of diverse proteins simultaneously. While this experimental technique (co-fractionation/mass spectrometry (CF/MS)) was originally developed to discover protein complexes from sets of co-eluting proteins (Havugimana et al, 2012; Kristensen et al, 2012; McWhite et al, 2020), here, we repurpose it as a technique to instead survey proteins for truncation variants at high throughput. For each protein, we generate a peptide heatmap that identifies where peptides unique to that protein appear across the fractions (Fig. 1A). Considering that a protein can elute at varying stages in a native fractionation due to participation in a protein complex or its ability to multimerize, we use peptide coverage as a tool to differentiate between truncated proteoforms and full-length proteins. A full-length protein will generally have peptide coverage across its entire length. In contrast, a truncated protein fragment will have peptide coverage solely at the N- or C-terminus of the protein. Fractionation thus allows proteoforms with different peptide coverage to be distinguished from one another.

While first denaturing proteins prior to separation would allow more faithful assessment of relative sizes, in principle, retaining native interactions and protein-protein associations might allow functions of short proteoforms to be more directly interrogated. Importantly, buffers, detergents, and separation conditions for CF/MS are chosen to maximize recovery of native, endogenous protein complexes with minimal disruption. The non-denaturing conditions that preserve protein interactions can additionally protect proteins from unfolding and non-specific degradation by cellular proteases, with degradation additionally prevented by the addition of protease inhibitors to cell lysis buffers.

A set of proteins that consistently co-elute across multiple different biochemical separations can be robust evidence that those proteins interact stably. Based on this principle and the corresponding freedom from specialized reagents such as recombinant

tags, antibodies, or isotope labeling, CF-MS has been broadly applied to highly diverse samples in order to discover stably interacting proteins across multiple species and kingdoms of life (Havugimana et al, 2012; Kristensen et al, 2012; Wan et al, 2015; Aryal et al, 2017; McBride et al, 2019; McWhite et al, 2020; Liebeskind et al, 2020; Pourhaghighi et al, 2020; Pang et al, 2020; Skinnider et al, 2021; Skinnider and Foster, 2021). Importantly, as CF/MS typically employs bottom-up mass spectrometry, the elution behaviors for individual (usually tryptic) peptides are recorded. This, in turn, enables their associated proteoforms to be distinguished, and in principle their native, co-eluting interaction partners to be simultaneously assayed.

Numerous CF/MS experiments spanning a variety of species are already available in the public domain to be analyzed post-hoc for truncated proteoforms, providing a rich resource for discovering new proteoforms and assessing their conservation across the tree of life. Beyond allowing the discovery of truncated proteins, these datasets should also allow the discovery of protein interactions between those fragments and other proteins.

Here, we surveyed elution patterns of proteins in high-throughput CF/MS fractionation experiments to detect short proteoforms that are found in high abundance, searching especially for those that do not correspond to annotated splice variants. We show that large-scale bottom-up mass spectrometry can identify truncated proteoforms without limitations on longer protein length. We recover both known and novel short proteoforms in nonapoptotic human cell culture and primary red blood cells, and in two plant species, *Arabidopsis thaliana* and the single-celled green alga *Chlamydomonas reinhardtii*. We observed general trends among short proteoforms with respect to processing intrinsically disordered tails, and we identified several novel proteoforms that have been shown recombinantly to have distinct function from the full-length protein, but have not previously been shown to exist endogenously in cells. Finally, a subset of short variants are found conserved across both the human and plant lineages, suggesting that they may date to their last common ancestor 1.5 BYA or potentially older.

# Results

## Co-fractionation/mass spectrometry reveals alternative proteoforms

We sought to identify short proteoforms of proteins in a systematic fashion, to determine their prevalence and evolutionary conservation, and to identify potential mechanisms for their biological production. Using the CF/MS method and taking advantage of abundant previously collected datasets as well as new ones, we developed an analytical approach to identify proteoforms with distinct sizes, charges, and peptide compositions (Fig. 1A). By examining the elution profile for each detected tryptic peptide in a protein, and ordering these elutions from the N- to C-terminus, we can detect proteins that are shorter than their expected full length.

Then, by comparing these observations to transcript isoforms, we could provide support for alternative generative mechanisms (transcription start/stop, splicing, or proteolytic (Fig. 1B)). We specifically target limited/regulated proteolysis, and not degradative proteolysis (Fig. 1C). As CF/MS is easily applied across species and

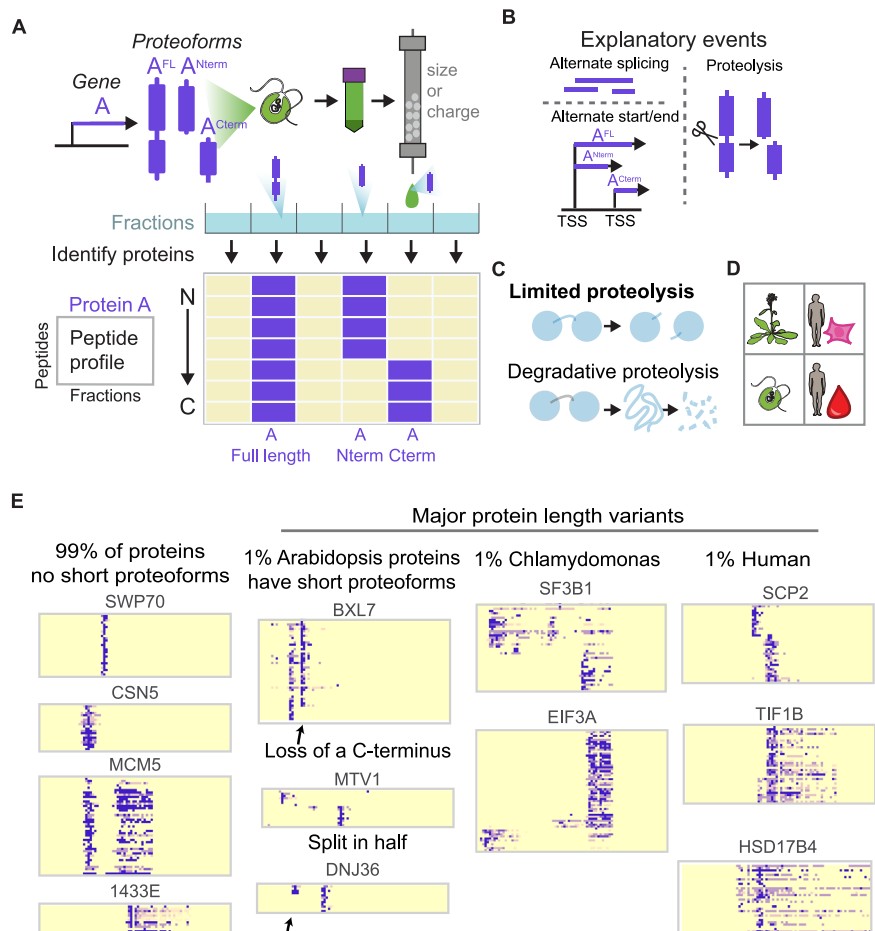

**Figure 1. Alternative proteoforms are evident in an examination of peptide elution profiles from co-fractionation / mass spectrometry experiments.**

(A) One gene can produce multiple proteoforms. To detect these proteoforms, cells are lysed, and the native lysate biochemically fractionated by, e.g., ion exchange or size-exclusion chromatography, with proteins in each fraction subsequently identified and quantified by protein mass spectrometry. Distinct proteoforms can be visualized and identified by plotting the elution patterns of individual peptides ordered from the N- to the C- terminus. A full-length protein has peptides covering the whole sequence of the protein, while truncated proteoforms will only have peptides from one terminus. These proteoforms can derive from alternate splicing, alternate start/end transcription sites, or proteolysis. (B) Truncated peptides can be derived from alternative splicing, proteolysis, and alternative start/end points on mRNA. (C) Limited proteolysis involves a specific cleavage point in the protein chain, in contrast to non-specific degradative proteolysis. (D) Species and samples analyzed in this study. (E) While a large majority of proteins show only a single full-length proteoform in these datasets, roughly 1% of proteins exhibit additional, shorter proteoforms.

cell types, we could additionally use a comparative proteomics approach to discern if the processing events were evolutionarily conserved.

We considered the following datasets appropriate to address this question: We selected two HEK cell size-exclusion chromatography CF/MS experiments from (Mallam et al, 2019), one of which was treated with ribonuclease (RNAse) to help identify ribonucleoprotein complexes (Fig. 1D), and we additionally report a new HEK cell ion-exclusion chromatography CF/MS experiment. We also included four fractionations of human hemolysate, containing soluble proteins from red blood cells (RBC) (Sae-Lee et al, 2022), as these cells are enucleated and do not translate protein. Therefore, protein variants in this cell type must be observed rather than inferred from mRNA isoforms. To search for conserved proteoforms, we additionally analyzed three *Arabidopsis* and three *Chlamydomonas* size and ion exchange separations from (McWhite

et al, 2020). Overall, these data capture mass spectrometry evidence for 7,028,432 unique tryptic peptides from 13,502 proteins across 16 native biochemical fractions. Table EV1 describes the features of these experiments.

As expected, the vast majority of proteins elute in a consistent manner, with peptides from the whole length of the protein found in the same biochemical fractions, which indicates that the whole protein traveled intact through the column prior to trypsin digestion to prepare the samples for mass spectrometry (Fig. 1E). However, a small subset of observed proteins (~1%) exhibit distinctive patterns, where certain biochemical fractions contain only peptides mapping to a specific end of the protein, which indicates that multiple proteoforms existed in the lysate. Importantly, detecting a protein at multiple points in the fraction is not evidence for the existence of a truncation variant, as CF/MS preserves protein complexes and multimeric assemblies. For

example, MCM5 elutes at two points, likely corresponding to the larger assembled MCM complex and smaller monomeric MCM5. At each position, peptides cover the full length of the MCM5 sequence, showing that a full-length MCM5 was detected at each position in the fractionation. In contrast, while BXL7 also elutes at two points in the fractionation, it shows evidence for a truncation variant. While the first observation of BXL7 in the fractionation has peptides spanning the protein's full length, the second observation is missing peptides corresponding to the C-terminus of the protein.

## Identifying, scoring, and prioritizing candidate protein fragments

As each CF/MS experiment measures thousands of proteins, we developed a quantitative score to prioritize proteins exhibiting alternative proteoforms. We manually assembled a validation set of proteins with suggestive peptide elution profiles and then used these cases to develop a heuristic score to prioritize proteins that may exhibit short proteoforms (Fig. 2A).

Briefly, proteins tend to elute across chromatographic fractions in distinct peak(s) in which multiple peaks suggest the existence of proteoforms or intact proteins eluting with different binding partners. Therefore, we model a protein's elution profile across the fractions as a Gaussian mixture model. Then, for each modeled peak, we consider the set of associated peptides' abundances and compute a *terminal bias* score summarizing the weight of evidence for a short proteoform eluting at that chromatographic position based on the sampling bias of peptides to either side of that position in the protein:

$$terminal\ bias = \max_i(\log_2[(observed_{N,i}/observable_{N,i})/(observed_{C,i}/observable_{C,i})])$$

where $observed_{N,i}$ and $observed_{C,i}$ are the counts of unique peptides observed under that modeled Gaussian peak to the N- and C-terminal sides, respectively, of the potential proteoform end position $i$, and $observable_{N,i}$ and $observable_{C,i}$ are the respective counts of unique peptides to the N- and C-terminal sides of the potential end position $i$ that were observed for that protein across *all* fractions. The *terminal bias* score for a modeled peak conveys the strength of each candidate proteoform, as true truncation proteoforms will be missing observable peptides from one terminus or the other. We additionally score each peak using a Fisher's exact test on contingency tables of observed and observable peptides, applied to each candidate proteoform end position $i$ in a similar manner to the terminal bias score. We then take the position $i$ with the Benjamini–Hochberg adjusted minimum p-value to prioritize a peptide profile for manual examination (Fig. 2A), under the intuition this p-value is evidence of a significant peptide distribution difference between observed peptide distribution at the N- and C-termini of a modeled peak. We evaluated the discriminatory power of the terminus bias score and Fisher's exact tests by assessing a set of 278 proteins whose elution profiles were manually examined and found to contain one or more very clear candidate short proteoforms. As plotted in Fig. 2B, both scores perform extremely well at prioritizing protein elutions with peaks corresponding to

apparently truncated proteoforms, as compared to 1000 randomly selected proteins. While the terminal bias score and minimum p-value largely agree for a given protein elution profile (Fig. EV1), each score prioritizes some proteins with interesting profiles that the other misses. This motivates the development of a future improved scoring approach to prioritize proteins with interesting peptide profiles.

For this same set, for proteins with terminal bias scores over 2, we observe a precision of 90.7 and recall of 77.3, and a false discovery rate of 9.3% (Fig. 2C). We then verified high-scoring candidate proteoforms with a terminal bias score above 2 or minimum p-value less than $1 \times 10^{-6}$ visually, and for each proteoform added occurrences in other experiments, even if they did not reach either threshold (Fig. 2D). The full set of proteoforms and annotated sequence coverage is provided in Dataset EV1. There is a clear abundance and peptide coverage dependency in our ability to detect proteoforms. Hand-selected proteoforms were generally much more abundant than background, and score-detected proteoforms had a minimum abundance of ten peptides. Low abundance proteoforms were only detectable if the proteoform was observed to occur at higher abundance in a different experiment (Fig. 2D). The number of observed proteoforms roughly correlates with sampling depth of individual experiments (Fig. 2E). Though many proteins satisfy criteria for abundance and peptide coverage, the total number of proteins with detected short proteoforms is relatively low, ranging from 0 to 100 per fractionation experiment, where typically over 5000 proteins are confidently identified. While native size-exclusion chromatography conflates molecular weight and hydrodynamic shape, and bound interaction partners will alter elution positions, we nonetheless observe that the full-length proteoform elutes earlier than the short proteoform for 73% of the cases, consistent with the expectation for having a larger molecular weight.

Importantly, not every gap in sequence coverage corresponds to an alternative proteoform. A limitation of our approach is that multiple peptides from a protein must be observed to create robust peptide profiles. The protein must be sufficiently high in abundance, and its component peptides detected with some consistency. Our investigation of short stretches of consecutive undetected peptides (<200 amino acids) has thus far entirely led to peptides that are poorly observable. These can correspond to hundreds of undetected amino acids flanked by regions of well-detected peptides. Many of these gaps may correspond to regions of poorly observable (non-"proteotypic") peptides in LC/MS due to poor ionizability, or be in a region of few cleavable peptides, for example, stretches with few lysines or arginines for trypsin digestion (Mallick et al, 2007; Lu et al, 2007; Qeli et al, 2014), or simply not contain unique peptides, complicating their assignment in mass spectrometry reference database matching.

Nonetheless, we find that a minority of protein-coding genes produce short proteoforms that are both highly abundant and of sufficiently different size/charge variation from the long form to cause distinguishable elution behavior. The robust observations of these short proteoforms in nonapoptotic cells suggest that these short proteoforms are stably maintained. These length variants can arise from alternative splicing, alternate transcriptional start/stop sites, or limited proteolysis. In the following sections, we highlight proteins that reflect diverse features and origins of truncated

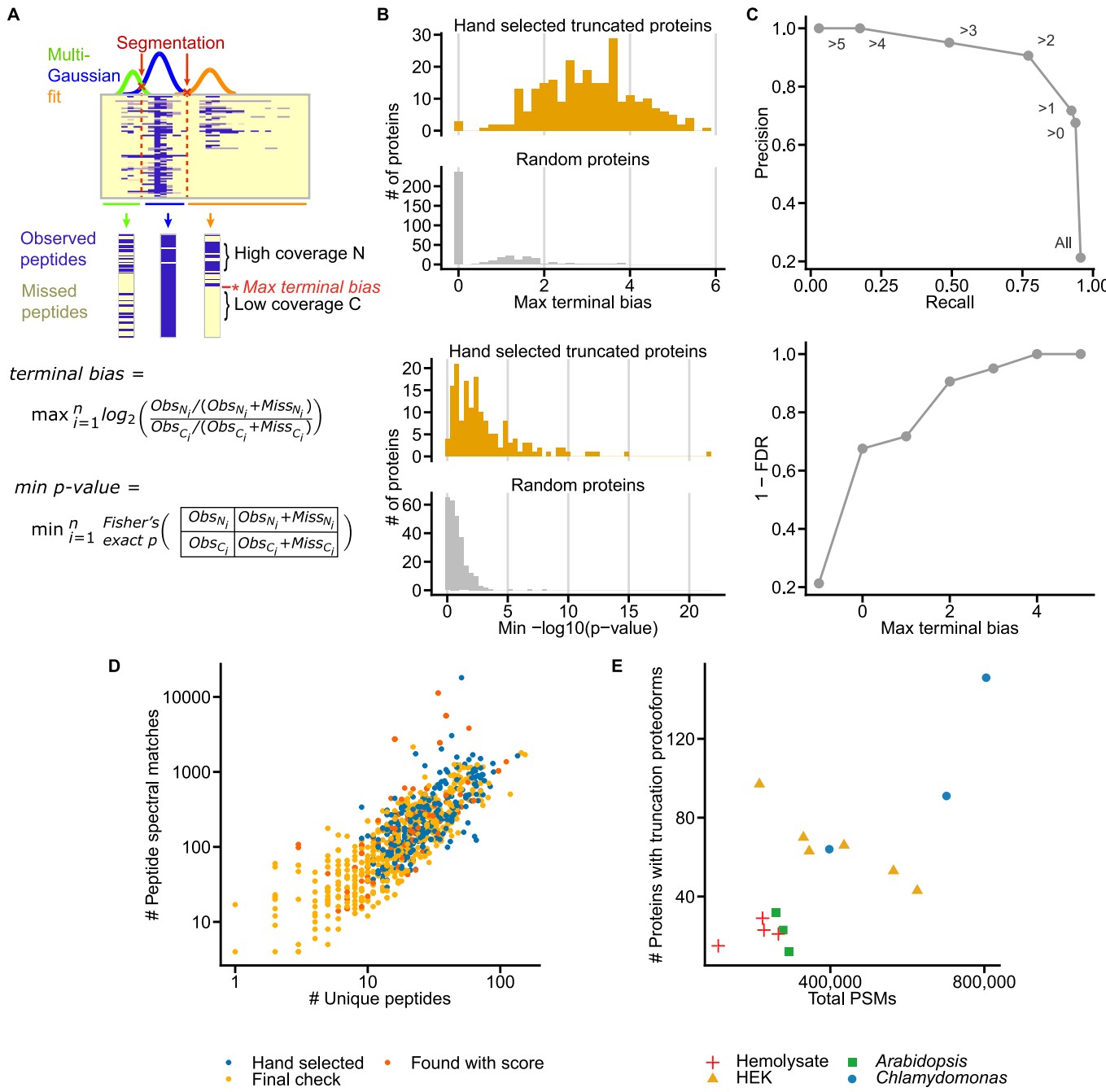

**Figure 2. Automating the detection of alternative proteoforms in plant and human CF/MS experiments.**

(**A**) Multiple Gaussians are fit to a protein's peptide elution profile. The intersections of these Gaussians are used to segment the elution profile. For each segment, we calculate a terminal bias and Fisher's exact *P* value at each peptide. For each protein, we save the maximum terminal bias and minimum p-value, which corresponds to the start or truncation point of a proteoform. (**B**) Proteins with apparent short proteoforms have higher max terminal bias scores (top) and lower minimum P-values (bottom) than randomly selected proteins. (**C**) Precision recall curve (top) and False discovery rates (bottom) of a positive set of 275 hand-selected proteins with apparent short proteoforms and a negative set of 1000 randomly selected proteins at different threshold terminal bias scores. (**D**) Compared to the background distribution of unique peptides per protein and peptide spectral matches, proteins for which we are able to detect proteoforms are relatively abundant. The terminal bias score prioritizes some proteins with lower abundance than our hand-selected proteoforms. In a final visual check, for some experiments, we identified a short proteoform at low abundance if the same protein had a more confident proteoform in a different experiment. (**E**) Number of proteoforms identified per experiment depends on the number of unique peptides identified in the experiment.

proteoforms, specifically focusing on "positive control" truncation variants that have been previously described in the literature, proteoform-specific protein interactions, and proteoforms with potentially biologically interesting roles.

## Short proteoform boundaries occur preferentially outside of domains

Although pinpointing the exact start or truncation point of a short proteoform depends on peptide coverage in the mass spectrometry experiment, we observed that these positions tend to occur outside well-defined protein domains. For example, for 214 observed

human proteoforms, 72% of their short proteoform boundaries fall outside protein domains (measured using InterPro (Blum et al, 2021)), while these proteins are on average 54% non-domain. This finding follows the observation by Hans Neurath that limited proteolysis occurs at protein "hinges and fringes" outside of domains (Hans Neurath, 1980).

The erythrocyte-specific proteoform of calpastatin is known to lack an N-terminal L-domain and the first of four inhibitory domains (Takano et al, 1986). We observe this short proteoform of calpastatin across human hemolysate experiments, and the expected full-length version in human HEK cell experiments (Fig. 3A). Note that the *y* axis does not directly scale to position

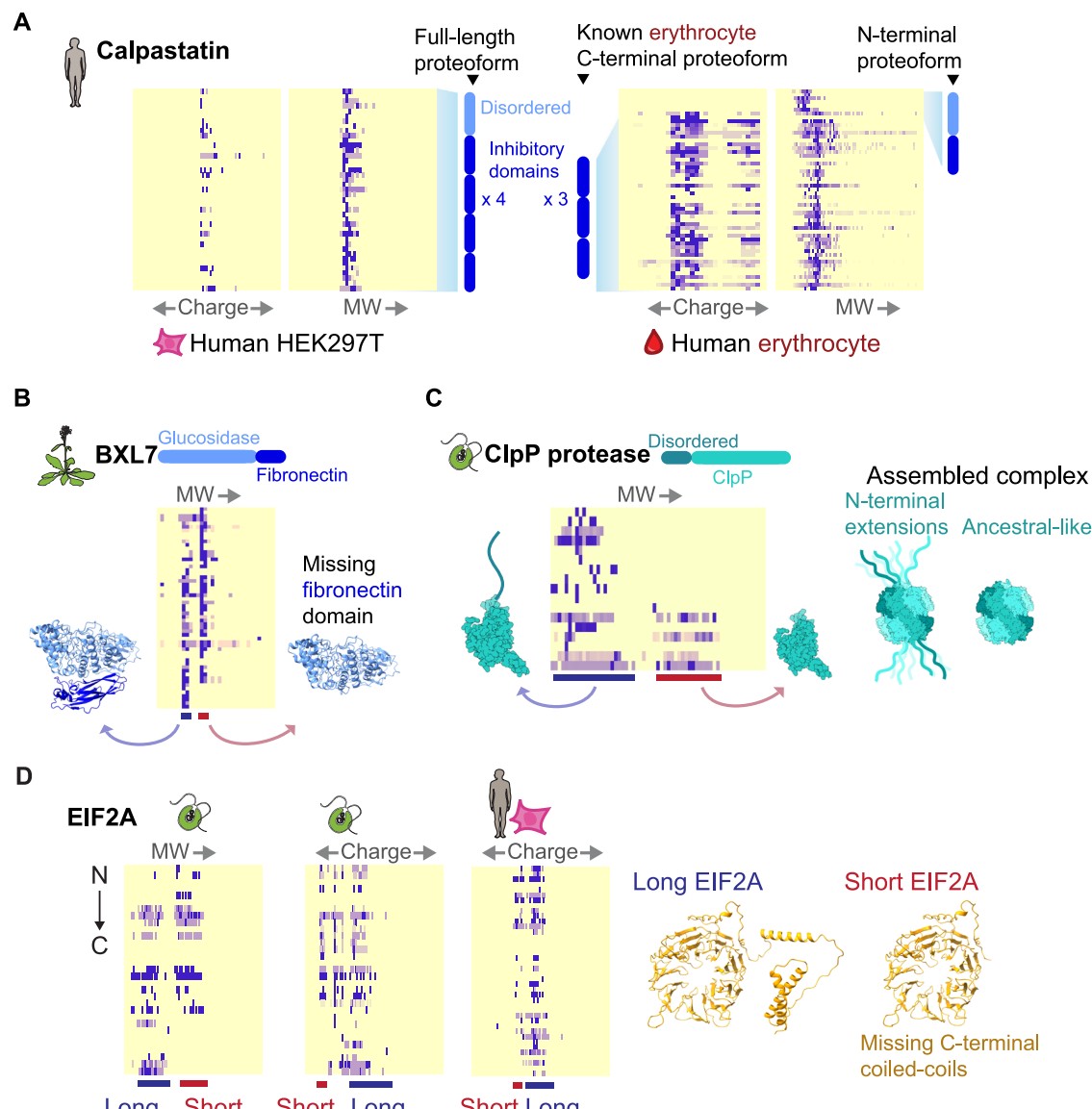

**Figure 3. Short proteoform boundaries occur preferentially outside of domains.**

(A) Cell-type-specific proteoforms of calpastatin. In HEK293T cells, we observe a full-length proteoform of calpastatin containing all domains, while in hemolysate we observe the erythrocyte-specific proteoform containing only the final three inhibitory domains. Unexpectedly, we observe an N-terminal proteoform of calpastatin in hemolysate. (B) In *Arabidopsis* we observe two proteoforms of BXL7, one with, and one without the c-terminal fibronectin domain. (C) In *Chlamydomonas*, we observe two forms of the ClpP protease, one with, and one without an essential N-terminal disordered extension. (D) In *Chlamydomonas*, and potentially humans, we observe a form of EIF2A with and without C-terminal alpha coils.

along the protein; rather, it indicates consecutively ordered tryptic peptides.

Interestingly, we observe an N-terminal fragment of calpastatin in hemolysate corresponding to the lost domains of the erythrocyte-specific proteoform. This start/end point occurs between the first and second inhibitory domains. In hemolysate, we also observe candidate proteoforms corresponding to the C-terminal regulatory domain of Ankyrin 1 (Hall and Bennett, 1987) and a proteoform corresponding to the stable C-terminal domain of Band 4.1 protein (Scott et al, 2001). While the regulatory domain of Ankyrin 1 has not been observed in vivo, its loss increases the vesicle affinity of Ankyrin 1 by eightfold (Hall and Bennett, 1987). In addition, proteoforms identified in our method appear to be complementary to proteoforms discovered by top-down proteomics. We compared our hemolysate proteoforms to the top-down Blood Protein Atlas erythrocyte proteoforms. While each method offers the opportunity to discover legitimate proteoforms, the above three proteoforms of calpastatin, Ankyrin 1, and Band 4.1 are exclusively found in our experiments, and there is otherwise no overlap between identified proteoforms.

Another particularly clear example of this apparent domain boundary trend can be seen in *Arabidopsis* beta-D-xylosidase 7 (BXL7) (Fig. 3B). The left (i.e., higher molecular weight) peak is composed of peptides from across the entire length of the protein. The right (lower molecular weight) peak contains peptides exclusively from the N-terminal glucosidase domain, and is missing all peptides from the C-terminal Ig-like fibronectin domain. Beta-D-xylosidases degrade xylan sugars, and variably contain a terminal Ig-like domain. These two proteoforms suggest that BXL7 performs dual roles, one dependent on its fibronectin extension, and one not. As the BXL7 gene has only one annotated transcript isoform (Lamesch et al, 2012), this event may result from proteolysis or an as-yet undetermined new isoform.

Similarly, in the *Chlamydomonas* ClpP protease CLPP1, an in-frame viral DNA insertion is known to have added a long disordered tail to the encoded ClpP1 protein (Derrien et al, 2009), which likely extends from the assembled ClpP protease complex (Fig. 3C). This tail is known to be proteolytically cleaved at the domain boundary to produce both heavy and light forms of the ClpP1 protein, both of which are essential (Derrien et al, 2009). We observe both heavy and light forms of the *Chlamydomonas* ClpP

protease (Fig. 3C). In the same way, we observe a long and short form of EIF2A in humans and *Chlamydomonas*, where the short form lacks a C-terminal coiled-coil region with no described function (Fig. 3D).

## Alternative transcript-derived proteoforms

By comparing truncated proteoforms' annotated mRNA transcripts, we can assign a potential source for each proteoform, i.e., whether it is more likely derived from a transcript variant or from limited proteolysis. 51.7% of reviewed human proteins in the Uniprot database have RNA isoform variants annotated. However, the extent to which these isoforms correspond to broadly expressed proteins is unclear (Aebersold et al, 2018). In our data, we find that 9 truncated proteoforms in humans (out of 142 total identified) are consistent with annotated transcription variants (Table 1). These transcript variants derive from either alternate splicing or alternate transcription start/stop sites. All these truncations fall between domains, suggesting that alternate splice isoforms that end mid-domain may generally not manifest into abundant protein.

For example, we observe both a full-length proteoform corresponding to the full trifunctional purine biosynthesis enzyme PUR2 (GARS-AIRS-GART) and a second N-terminal proteoform corresponding to only the first enzyme, GARS (Fig. 4A). A short transcript isoform of PUR2 (Henikoff et al, 1983) contains only the GARS enzyme and uses the same 10 exons as the full-length isoform, but is terminated by an intronic polyadenylation signal between exons 11 and 12 (Kan and Moran, 1995). We observe the trifunctional GARS-AIRS-GART and monofunctional GARS proteoforms eluting as distinct peaks.

While we initially targeted truncation variants, we do observe a clear case of a proteoform with a missing internal domain (Fig. 4B). We observe the two major forms of Nuclear Autoantigenic Sperm Protein (NASP), the full-length t-NASP and the shorter s-NASP proteoforms (Richardson et al, 2000). S-NASP is produced by alternate splicing of the NASP transcript, where internal exons are spliced out. An examination of the proteomic evidence for NASP confirms the presence of two proteoforms, one involving peptides from the entire NASP sequence and the second missing a large internal segment. Western blots from the Human Protein Atlas (Thul et al, 2017) independently confirm that both the long and

**Table 1. Correspondence of observed short proteoforms with known human isoforms.**

| ProteinID | Term | Observed coverage | Isoform coverage | Uniprot Isoform ID | Observed experiments |
|---|---|---|---|---|---|
| sp|P22307 | NLTP_HUMAN | C | 405–547 | 405–547 | SCP2 | HEK_SEC2, HEKR_SEC2 |
| sp|P22102| PUR2_HUMAN | N | 5–433 | 1–433 | Short | HEK_IEX1, HEK_IEX2, HEK_SEC1, HEK_SEC2, HEKR_SEC1 |
| sp|Q5JSH3 | WDR44_HUMAN | C | 525–886 | 474–913 | 3 | HEK_IEX1 |
| sp|P20290 | BTF3_HUMAN | C | 80–206 | 45–206 | 2 | HEK_IEX1, HEKR_SEC1 |
| sp|P46108 | CRK_HUMAN | N | 21–179 | 1–204 | Crk-I | HEMO_IEX1, HEMO_IEX2, HEMO_SEC1, HEMO_SEC2 |
| sp|P35249 | RFC4_HUMAN | N | 7–232 | 1–303 | 2 | HEK_IEX2, HEK_SEC1 |
| sp|Q4VCS5 | AMOT_HUMAN | C | 413–856 | 410–1084 | 2 | HEK_IEX1, HEK_SEC1, HEKR_SEC1 |
| sp|Q71RC2 | LARP4_HUMAN | N | 139–428 | 1–445 | 7 | HEKR_SEC1 |
| sp|A0MZ66 | SHOT1_HUMAN | C | 467–593 | 413–631 | 7 | HEK_IEX2 |

Observed coverage for N-terminal proteoforms spans from the N-terminus to the last observed amino acid. Observed coverage for C-terminal proteoforms spans from the first observed amino acid to the C-terminus.

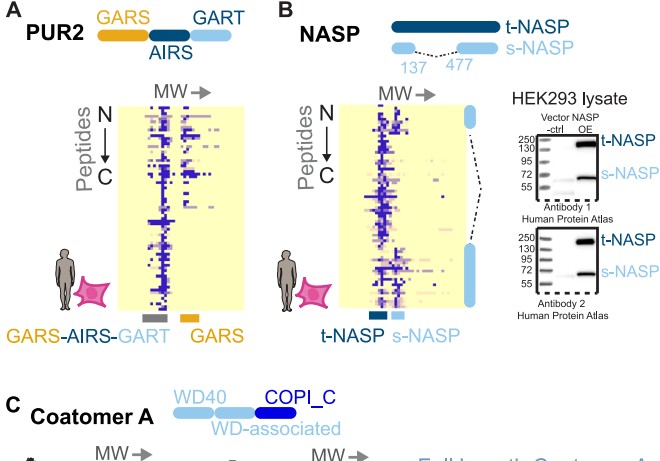

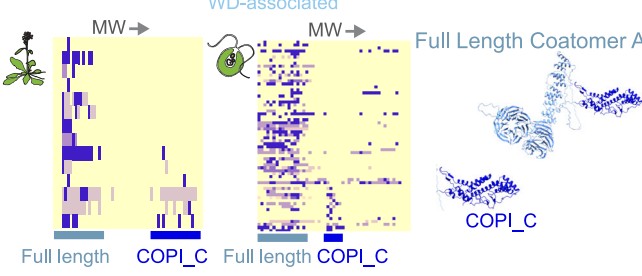

**Figure 4. Known and novel transcript-directed generation of short proteoforms.**

(A) The *GART* gene encodes both a trifunctional enzyme GARS-AIRS-GART and the mono-enzyme GARS. The short form derives from an early transcription stop site. We detect both forms. (B) Two known proteoforms of NASP, t-NASP and s-NASP, are detected in HEK293T elutions. The same proteoforms are detected with two independent antibodies in the Human Protein Atlas(V21). Images available at http://www.proteinatlas.org/ENSG00000132780-NASP/antibody. (C) Two major proteoforms are evident for the Coatomer A protein from *Chlamydomonas*, a full-length COPA and a C-terminal proteoform composed of only the COPI_C domain, consistent with mRNA isoforms corresponding to this novel short protein being found broadly across eukaryotes.

short forms exist in HEK cells (Fig. 4B). NASP is a rare example in our data of a protein with a distinct proteoform with clear internally spliced out exons, though our ability to detect this type of proteoform is highly limited by abundance and peptide coverage.

Finally, we highlight our discovery that a particular Coatomer A alternate transcript is translated into protein, and likely broadly conserved across eukaryotes. Two major proteoforms of Coatomer A are clearly distinguishable in both *Chlamydomonas* and *Arabidopsis*: a full-length COPA and a novel C-terminal fragment composed of only the COPI_C domain (Fig. 4C).

Unfortunately, the protein evidence is less clear in our human samples. The observed novel short COPI_C-only protein is consistent with an mRNA isoform covering only the COPI_C domain of COPA. The isoform matching the COPI_C proteoform (A0A3B3ITI7) is only one of 13 isoforms of COPA in humans in the UniProt database (UniProt Consortium, 2019).

## Many proteoforms likely derived from proteolysis

In the absence of already known transcript isoforms, the remainder of human truncation proteoforms likely either arise from uncharacterized mRNA isoforms or proteolysis, and potentially in some cases, a combination of both events. For example, while we observed truncated proteoforms of BTF3 and CTSC that match mRNA isoforms, these proteins also each show a second truncated proteoform that incorporates sequences not present in the known transcript isoforms (Fig. 5A,B), suggesting that these fragments likely derive from limited proteolysis. While the C-terminal fragment of BTF3 is consistent with the isoform BTF3b, the presence of the N-terminal fragment suggests that these proteoforms derive from proteolysis (Fig. 5A).

Though it can be difficult to prove that a proteoform does not derive from mRNA isoforms, certain proteoforms that we detect are either known to derive from proteolysis, or are similar to known proteolyzed proteins. We observe multiple cases of known limited proteolysis in our data. For example, the observed N-terminal fragment of HSD17B4 matches a fragment produced by a known proteolysis event (Fig. 1E) (Okumotoet al, 2011). We can bring to bear additional evidence from the set of proteoforms observed. In particular, when non-overlapping N- and C-terminal fragments are observed, we consider it likely that the fragments derive from cleavage. For example, human erythrocyte calpastatin exhibits non-overlapping N- and C-terminal fragments, suggesting that these fragments derive from proteolysis (Fig. 3A).

Endogenously, limited proteolysis is generally a regulatory event. Proteins that undergo a conformational change upon activation are natural candidates for proteolytic processing, as conformational change can expose hidden cleavage sites. For example, the protease Cathepsin is inactive until proteolysis of its N-terminal propeptide (Butler et al, 2019), when binding of PGRN exposes a hidden cleavage site to proteases (Fig. 5B). We see the processed form of cathepsin in humans, *Arabidopsis*, and *Chlamydomonas*, and also capture the cathepsin propeptide in humans (Fig. 5B).

The functional purpose of regulated proteolysis is not always so clear. Translocon at the Outer envelope membrane of Chloroplasts 159 (TOC159) contains an N-terminal highly acidic domain (A-Domain), a GTPase domain (G-domain), and a membrane anchor (M-domain).

Originally thought to be a degradation product of protein extraction, the A-domain has now been shown to likely exist as an independent proteoform in vivo, and cleaved by an unknown protease (Agne et al, 2010). We observe three proteoforms of TOC159 (1) the A-domain alone, (2) the A-domain with the GTPase domain, and (3) a full-length proteoform in *Arabidopsis* (Fig. 5C). For proteins known to be involved in stable complexes, short proteoforms tend to either co-elute with the same interaction partners as the full-length protein, or not co-elute. This appears to be related to macromolecular structure. The full-length proteoform co-elutes with TOC75-3, another member of the TOC complex. These biochemical fractions show enrichment for peptides from the membrane domain, potentially indicating the presence of a membrane anchor proteoform. In contrast, neither of the N-terminal short proteoforms containing the A-domain co-elute with TOC75-3. Our observation supports endogenous occurrences of both the A-domain alone, and a candidate second proteoform containing the A-domain and the GTPase domain, which both do not remain attached to the TOC complex.

Ligand binding to the NOTCH membrane receptors induces a conformational change that exposes a hidden metalloproteinase

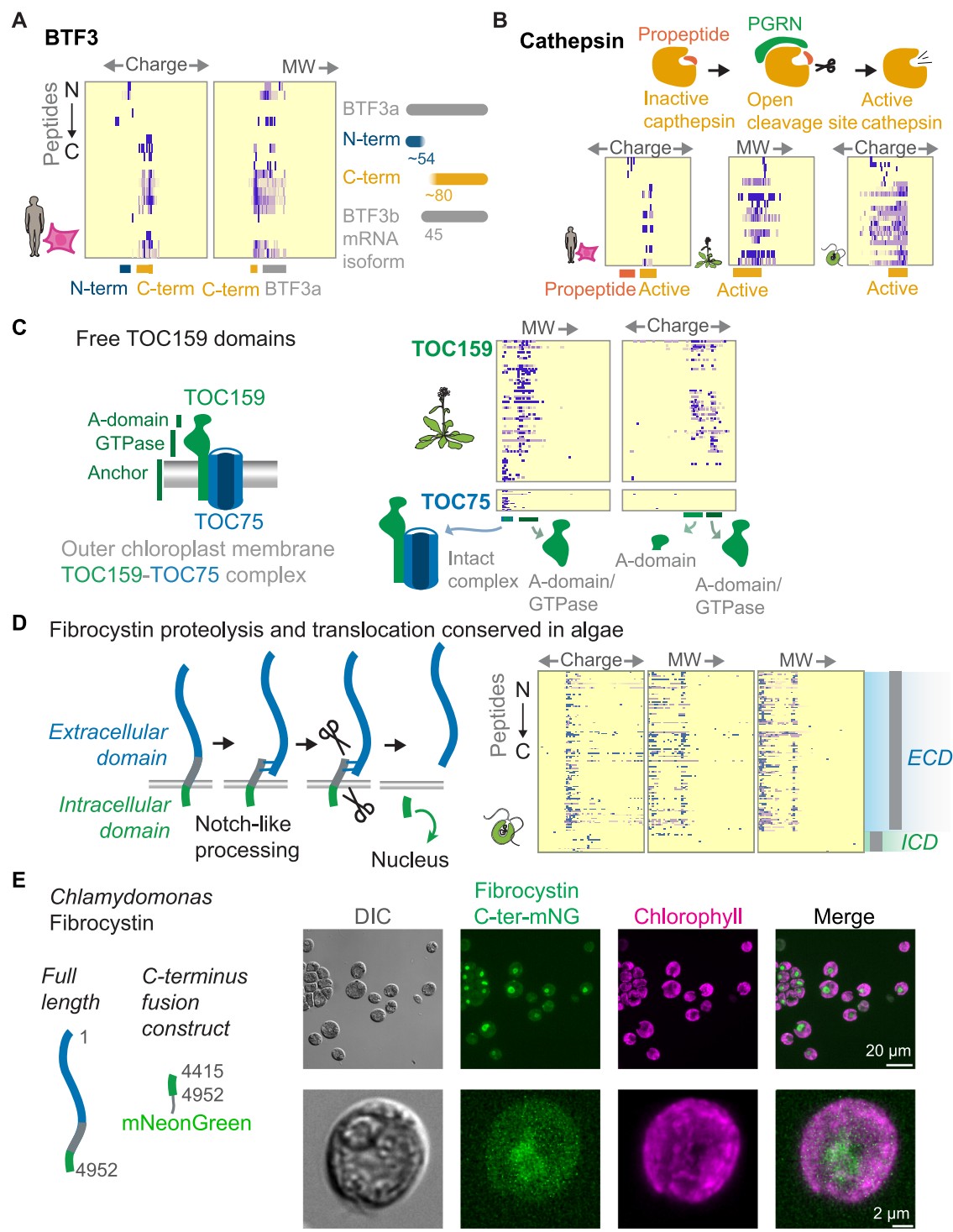

site, allowing cleavage and release of the cytoplasmic portion (Gordon et al, 2007). For signaling molecules, proteolysis can allow translocation of a signal after activation, such as is the case for Notch-like proteins. Fibrocystin is one such Notch-like protein in animals associated with polycystic kidney disease that is known to be first translated as a membrane protein and then proteolytically processed—probably by a proprotein convertase and ADAM metalloproteinase disintegrins—into N and C-terminal fragments

that translocate away from the membrane into other compartments, with the C-terminus trafficking into the nucleus (Hiesberger et al, 2006; Kaimori et al, 2007; Follit et al, 2010). Similar Notch-like processing activities have not yet been characterized in green algae and plants. Nonetheless, *Chlamydomonas* has a fibrocystin/ PKHD1-like ortholog (Cre07.g340450/A0A2K3DKH4) that possesses a putative signal peptide and one transmembrane domain (Käll et al, 2007), has been confirmed to be N-glycosylated

◀ **Figure 5.   Numerous proteoforms likely derive from proteolysis.**

(**A**) In humans, we observe N- and C-terminal proteoforms for the transcription factor BTF3. While a C-terminal mRNA isoform exists, the presence of the N-terminal proteoform suggests that these proteoforms derive from proteolysis. (**B**) In humans, *Arabidopsis*, and *Chlamydomonas* we observe activated cathepsin, where the N-terminal propeptide has been removed by the interaction with PGRM. In humans, we observe the propeptide alone along with the activated form. (**C**) TOC proteins form a complex at the chloroplast membrane. We observe TOC159 in three forms, one full-length proteoform in complex with the TOC membrane anchor TOC75, one N-terminal proteoform containing the A-domain and the GTPase, and one N-terminal proteoform containing only the A-domain. Neither of the N-terminal proteoforms co-elute with TOC75 showing that these proteoforms have disassociated from the rest of the TOC complex. (**D, E**) Fibrocystin/PKHD1 undergoes proteolysis at two positions, one in the N-terminal extracellular domain, which undergoes NOTCH-like cleavage, and one at the intracellular C-terminal domain which releases the C-terminus to go to the nucleus. While this processing is known in animals, it was not known to occur in green algae, but the observed proteoforms (at right) support fibrocystin cleavage in *Chlamydomonas*. (**E**) A *Chlamydomonas* fibrocystin C-terminal proteoform can localize to the nucleus. A truncated fibrocystin sequence (amino-acid positions 4415–4952) fused to mNeonGreen was expressed from the constitutive *PSAD* promoter.

(Mathieu-Rivet et al, 2013), and thus is most likely membrane-localized in a manner similar to its animal orthologs (Vagin et al, 2009). We observe that Cre07.g340450 exhibits short proteoforms that correspond to the mammalian ortholog's proteolytic fragments: a large N-terminal proteoform containing the transmembrane domain and the much shorter C-terminal proteoform (Fig. 5D). Due to the large size of the full-length protein (606 kD), we opted to validate the observed C-terminal proteoform with a recombinant tag. The mNeonGreen tagged C-terminal proteoform is indeed localized to the nucleus, supporting the occurrence of a Notch-like signaling process in *Chlamydomonas* similar to that observed in animals (Fig. 5E).

It is important to note that no processing mechanism is yet known in the plant case, and land plants seem to lack ADAM metalloproteinase disintegrins (Blum et al, 2021). However, the evolutionary conservation of the shorter proteoforms and the nuclear localization of the C-terminal construct in *Chlamydomonas* suggest the possibility that Notch-like proteolytic processing is conserved in some fashion in green algae.

## Isolated intrinsically disordered termini as stable short proteoforms

Many proteins contain an intrinsically disordered terminus (IDT) on one or both ends. Some IDTs may exert an entropic force that can shift protein conformation favorably, with a force proportional to the length of the disordered region (Keul et al, 2018). Other IDTs modify solubility, particularly in the case of phase-separated proteins (Elbaum-Garfinkle et al, 2015).

In contrast to our initial assumption that a lost disordered region would be immediately degraded, we detected multiple proteoforms composed entirely of a disordered terminal domain separate from the full-length protein. These IDT-only proteoforms have a consistent sequence coverage and length, suggesting regulated proteolysis or transcription, and not non-specific degradation by cellular proteases.

We detected these intact disordered termini of proteins either alone or alongside their full protein. A striking example of this can be seen with EIF3B, which is observed in its full-length form in human HEK cells, but for which we observed only the 113 amino acids of the N-terminal disordered region in RBC hemolysate (Fig. 6A). This distinct pattern would be missed in quantification that pools observed peptides. The unique detection of the EIF3B disordered terminus alone, in the absence of the full-length protein, suggests that this disorder-only proteoform is stable, long-lived (as

RBCs lack transcription and translation to regenerate fresh copies of the protein), and not an artifact of sample preparation.

In contrast to the EIF3B N-terminal proteoform, which is observed in both size exclusion and ion exchange chromatographic separations, there are other disordered termini which we only observed in ion exchange experiments. For EIF3A, another member of the EIF3 complex, we observed two distinct disordered C-terminal proteoforms in both *Arabidopsis* and *Chlamydomonas* ion exchange but not size-exclusion separations (Fig. 6B). In this case, we cannot rule out that these disordered termini were not lost during sample preparation. Regardless, they highlight the need to check if protein elution peaks contain full protein, or only portions thereof.

Most transcription factors (J Liu et al, 2006) and many other proteins involved in regulatory control of transcription and translation (Dyson and Wright, 2005) contain at least one disordered terminus. We frequently detect at least one free disordered terminus in abundant proteins with disordered termini. For example, for both the DDX21 helicase in humans and the RH38 helicase in *Arabidopsis*, short proteoforms corresponding to independent disordered N-termini are apparent (Fig. 6C). Other helicases we observed terminal proteoforms for are DDX6, DDX17, DDX46, DDX50, and DHX9. We additionally observed processing of multiple translation factors, including EIF4G1/2 and EIF3B in human data, IF-2 in *Arabidopsis* and *Chlamydomonas*, EIF2A in human and *Chlamydomonas* data, and EIF2B and EIF5B in all three species.

## Truncated proteoforms of translated-related proteins, including EIF5B and G3BP1, resemble enterovirus infection-induced processing

The disordered termini of numerous proteins involved in gene expression and translation also share another curious feature–they are known to be specifically cleaved by enteroviruses during infection, a modification which increases the viral load in infected cells (de Breyne et al, 2008). For example, viral proteolysis of the disordered terminus of EIF5B (de Breyne et al, 2008) causes a shift in the function of the protein, which allows increased viral replication.

Interestingly, in the absence of infection, we observed endogenous proteolysis of proteins known to be cleaved by enteroviruses, including DDX6 (Saeed et al, 2020), HNRPM (Jagdeo et al, 2015), EIF5B (de Breyne et al, 2008), G3BP1 (Zhang et al, 2018), and eIF4G1/2 (Etchison et al, 1982). Interestingly, all these proteins are known to phase-separate (You et al, 2020).

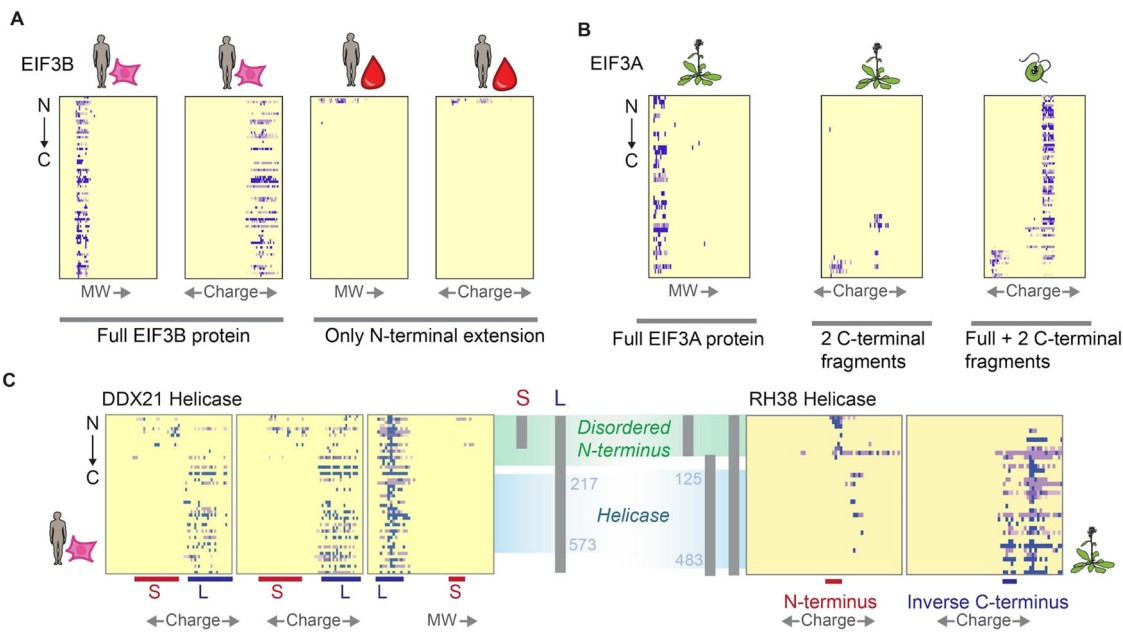

**Figure 6.  Numerous endogenous proteins exhibit loss of a disordered terminus corresponding to known enterovirus cleavage events.**

(A) While in HEK293 cells, we observe full-length EIF3B, in erythrocytes we only observe peptides from its short N-terminal disordered region. (B) We observe various C-terminal proteoforms of EIF3A in *Arabidopsis* and *Chlamydomonas*. (C) For proteins with a disordered terminus, we frequently observe either independent proteoforms of just the disordered portion, or loss of disordered termini.

## G3BP1 exhibits proteoform-specific protein interactions

Among the enterovirus-related proteolysis events, G3BP1 is known to be cleaved by enterovirus 3C protease cleavage, leading to decreased granule formation and increased viral replication (Zhang et al, 2018). G3BP1 has been recently shown to act as a conformational switch (Guillén-Boixet et al, 2020; Yang et al, 2020). We find a proteoform composed of the G3BP1 C-terminal RRM RNA-binding domain plus its C-terminal disordered region (Fig. 7A), which matches the proteoform produced by viral proteolysis. The remainder of the protein (i.e., the "other half" of our observed proteoform) has also been detected in an immuno-precipitation experiment of G3BP1 (Sanders et al, 2020). We observed that this short proteoform of G3BP1 tends to associate with a subset of stress granule proteins relative to the full-length G3BP1 (Fig. 7B). While viral proteases are known to disrupt stress granules by cleaving G3BP1, our data suggest that an unknown endogenous protease may cleave G3BP1 at a similar position (Fig. 7C). Enterovirus cleavage may push G3BP1 toward a particular state favorable to the viral replication, rather than merely inactivating the protein.

## A short proteoform of USP15 interacts specifically with quinone reductase 2, TANGO2, and α-hemoglobin-stabilizing protein in red blood cells

In order to more systematically search for rare cases of short proteoform-specific or proteoform-enriched protein interactions such as we observed for G3BP1, we scored elution profile similarity between all pairs of proteins or protein proteoforms, looking for those that tended to occur in the same fractions over multiple experiments. The highest scoring interaction, of similar strength to the consistent co-elution of proteasome components, was an N-terminal proteoform of Ubiquitin-specific protease 15 (USP15) specifically co-eluting with several other proteins: quinone reductase 2 (NQO2), Transport and Golgi organization protein 2 (TANGO2), and alpha-hemoglobin-stabilizing protein (AHSP). This interaction is strongest in red blood cell hemolysate (Fig. 7D), though an interaction between NQO2 and Nterm-USP15 is additionally observed in our HEK experiments.

Ubiquitin-specific protease 15 (USP15) is composed of three main domains: a uncharacterized <u>d</u>omain present in <u>u</u>biquitin-<u>s</u>pecific <u>p</u>roteases (DUSP), an ubiquitin-like domain (Ubl), and a ubiquitin-specific protease domain (USP). The DUSP-Ubl portion of USP15 enhances ubiquitin exchange (Clerici et al, 2014), and DUSP is not found in any non-USP domain architectures. Our complex contains mostly DUSP-Ubl, and little to none of the full-length USP15. Though this complex is intriguing, biological interpretation of this set of interacting proteins is difficult, especially given limited knowledge of component proteins roles in red blood cells.

Nonetheless, our data support multiple proteoform generation routes, as our observed USP15 N-terminal proteoform is similar in length to an mRNA isoform, (Q9Y4E8-4), differing only by a single terminal peptide. We additionally observe a C-terminal proteo-form, which does not correspond to mRNA transcripts. This suggests both a proteolytic and transcriptional route to produce the short form of USP15. Importantly, any proteoforms produced by the transcriptional route would necessarily have to be synthesized prior to red blood cell enucleation.

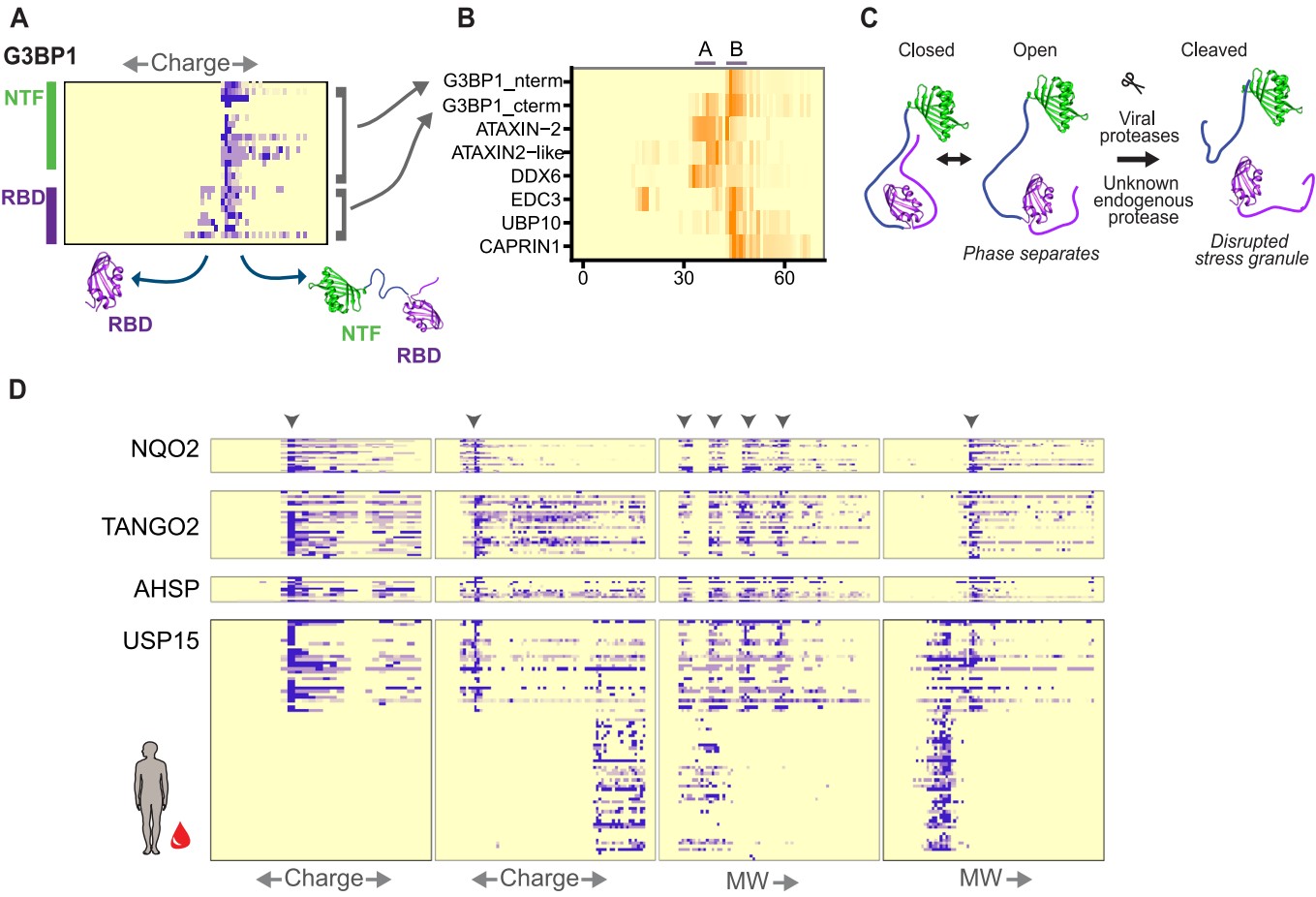

**Figure 7. Proteoform-specific protein interactions.**

(**A**) We observe both full-length and C-terminal RBD domain-only proteoforms of G3BP1. (**B**) The C-terminus of G3BP1 is enriched for interactions with ATAXIN2, ATAXIN2-like, and DDX6 proteins, relative to full-length G3BP1. (**C**) G3BP1 has open and closed conformations. Viral proteases cleave G3BP1, disrupting stress granule formation. An unknown endogenous protease may additionally disrupt stress granule formation via proteolysis of G3BP1. (**D**) We observe a protein complex of NQO2, TANGO2, AHSP, and the N-terminus of USP15 in erythrocytes. The C-terminus of USP15 and the full-length form of USP15 are not involved in the complex.

## Discussion

Biochemical diversity at the mRNA level derives mainly from alternative splicing and alternative transcription start and stop sites. While the contribution of mRNA variants to proteoform diversity is still actively being characterized and debated (Tress et al, 2017; Blencowe, 2017; Liu et al, 2017), our method does discover certain transcriptional proteoforms that substantially change charge or size characteristics of the protein. While our ability to detect these transcriptional proteoforms is dependent on high observed peptide coverage of a protein, examining the elution patterns of peptides from the N- and C-termini of proteins allows us to detect certain highly expressed short proteoforms without the need to consider rare isoform-specific peptides.

Indeed, we observed proteoforms clearly generated by both transcript-focused and proteolytic mechanisms, and in some cases, could distinguish alternative functions attributable to the short proteoforms based on their different protein interactions, as in Fig. 7. In general, our approach exhibits a strong bias for highly stable or abundant short proteoforms, but it seems likely that this

subset of abundant, stably maintained proteoforms would more often play functional roles in cells. For example, the cleaved C-terminal domain from the Fibrocystin-like protein, Cre07.g340450, in *Chlamydomonas* can translocate into the nucleus (Fig. 5D,E) while the full-length form of the protein is membrane-bound. This example shows the proteolytic process elicits cellular signals through compartmentalization of different proteoforms which in turn drives function.

There are multiple classes of protein functions where proteolysis has a functional effect beyond degradation (Fig. 8A). For example, in the case of zymogens, proteolysis releases a terminus which physically blocks a functional site, thus activating the enzyme (Neurath and Walsh, 1976). Multifunctional proteins may be separated into their component enzymes. Proteolysis also offers a quick route to disconnect structures in the cell that are connected by linker proteins. It may also quickly release phase-separated proteins from liquid droplets, or to allow both phase-separated and non-phase-separated populations of a protein. Proteolysis can allow the translocation of a signal peptide upon activation of a receptor. We find that several of our observed proteins with proteolytic fragments are known to exist as

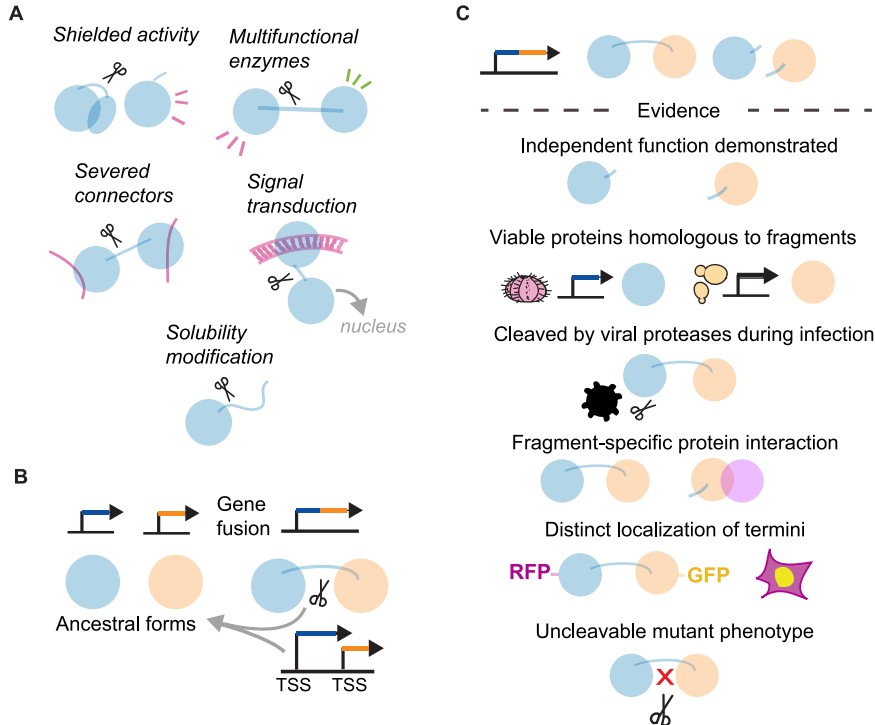

**Figure 8. Diverse modes of short proteoform generation and cellular function and evidence.**

(A) Proteolysis can play numerous functional roles beyond degradation in generating short proteoforms. (B) For the case of fusion proteins, the generation of short proteoforms may serve to re-create ancestral protein functionality while also benefiting from the larger fused protein. (C) Varying lines of evidence are suitable for learning the in vivo functions of a short proteoform.

conformational switches (e.g., G3BP1). While these switches are reversible, proteolysis may irreversibly lock a protein in one state or another.

Short proteoforms, whether produced by transcriptional or proteolytic processes, allow a gene to produce multiple distinct protein products, and this processing may be conserved evolutionarily. Strikingly, we observed several novel processing events to be conserved between humans and plants. These conserved short proteoforms included separation of disordered terminal domains from DDX helicases (Fig. 6C) and other translation-related proteins.

Proteolysis of signaling proteins likely often represents regulatory events. Though notch-like proteins are known to exist across plants, we demonstrate that notch-like proteolytic processing is conserved in green algae as well (Fig. 5D,E).

Another evolutionary aspect to the production of proteoforms is that when two genes fuse, it might be expected that the original genes' individual protein products will no longer be produced, in favor of one larger fused protein (Fig. 8B). However, generating short proteoforms of the fusion protein can result in retaining the ancestral biologically active proteins while also benefiting from the larger fused protein. Truncation of a protein has the ability to revert the protein product of a fused gene to that of one of the pre-fusion genes (Neurath, 1980). This appears to be the case for *Chlamydomonas* ClpP, which exhibits proteoforms with and without a lineage-specific extension (Fig. 3C). Limited proteolysis or alternate transcription of fused proteins creates an opportunity for

simultaneous retention of original function and neofunctionalization, without the need for gene duplication.

One notable finding from our survey was that short proteoforms are enriched in certain protein families, and that certain domains are more likely to exist independently or be lost.

Disordered domains were the most commonly lost domain, as for the case seen above for ClpP1 (Fig. 3C). Intrinsically disordered tails often regulate proteins they are attached to (Uversky, 2013). Disordered regions modulate phase separation behavior (Lin et al, 2017), and thus are key control points.

We particularly observed candidate independent disordered domains and loss of RNA recognition motifs (RRMs) from translation-related proteins, including multiple translation factors and helicases (Fig. 6A). Many of these proteins additionally are known to phase-separate, with phase separation modulated by disordered terminal domains (Ivanov et al, 2019; You et al, 2020). Interestingly, multiple of these proteins are additionally known to be cleaved by Enteroviruses at specific positions (Saeed et al, 2020). We see endogenous-specific proteolysis of these same translation-related proteins in human cells and plants, suggesting that Enteroviruses may mimic endogenous proteolysis events to induce a cell state that is more favorable to viral replication.

More generally, we observed numerous cases of proteoforms composed only of intrinsically disordered tails, independent from their parent proteins. Many wholly unstructured proteins perform functional roles in the cell, such as EPYC1 which mediates the phase separation of RuBisco in the algal pyrenoid body

(He et al, 2020). It is therefore not unreasonable to suggest that intrinsically disordered domains separated from their parent protein may also have a functional purpose. While disordered termini are particularly susceptible to degradation in in vitro purifications, in the cell, IDTs are generally stabilized by intermolecular interactions (Suskiewicz et al, 2011).

While it is known that the removal of disordered termini modifies protein properties, the observation of intrinsically disordered termini isolated from their parent protein suggests that some of these observed IDTs may play independent roles in the cell, rather than being immediately degraded. This is relevant to localization studies of proteins with fluorescent labels attached to disordered termini, as fluorescence may mark a truncated protein, and not exclusively the full-length version. In addition, we suggest a flag for protein data analysis for when only peptides from disordered termini are detected.

It is generally challenging to prove that an observed short proteoform exists in vivo, and has a functional role in the cell. This is due to the possibility of non-specific proteolysis or fragmentation any time cells are lysed, as is necessary for many biochemical assays. We suggest a series of sources of evidence for in vivo function of a short proteoform, whether observed from mRNA isoform, proteomic, or Western blot (Fig. 8C), from least strong to most strong evidence. One piece of evidence would be the demonstration of individual biological function of the short proteoform, exogenously expressed. In addition, the existence of a full-length protein corresponding to the short proteoform in either a different species or the same genome shows that the proteoform can have individual function in a cell. This follows the case of ClpP in *Chlamydomonas*, where the short proteoform of the protein matches the ClpP found in most other species (Fig. 3C). As enteroviruses modify the cellular environment through specific proteolysis of certain proteins, endogenous proteolysis at the same sites could be expected to be functional. Next, a shift in stable protein interactions suggests that a proteoform has a distinct function from its full-length version. In addition, if a short proteoform has distinct localization from its full-length protein, this indicates that there may be proteoform-specific transport, as in the case of Fibrocystin/Notch C-termini which are transported to the nucleus. Finally, the most rigorous evidence of a function of a specific proteolysis is to demonstrate that uncleavable variants of the protein expressed in a knock-out background cause a phenotype, or a loss of the production of the proteoform with the addition of protease inhibitors.

Peptide elution profiles provide a rich and systematic source of information about native proteoforms in complex biological systems. We surveyed proteins in CF/MS proteomics experiments to detect short proteoforms that are found in high abundance, and found that roughly 1% of proteins expressed short forms. Our method diverges from other approaches to detect proteoforms, as we are able to observe long proteoforms and additionally do not need to specifically enrich for terminal peptides. While we are limited by protein abundance and observed peptide coverage, new deep-learning methods that allow more peptides to be identified from proteomics experiments (Zeng et al, 2022; Ekvall et al, 2022) have the potential to increase the number of proteoforms detectable from a single experiment. In addition, while our heuristic score can prioritize proteins for manual inspection, confirming that a peptide profile contains a truncation variant and its approximate truncation point still requires visual inspection. While we have simplified this process with an interactive R Shiny application, hundreds of peptide profiles must still be visually screened. Introducing machine learning could allow more automated identification of truncated proteoforms from CF/MS experiments.

We recovered both known and novel short proteoforms in nonapoptotic human cell culture and blood and in two plant species. Many did not correspond to annotated splice variants and thus are likely to represent proteolytically generated proteoforms. A subset of short variants were found conserved across both lineages, suggesting that they may date to their last common ancestor 1.5 BYA or potentially older. We observed frequent truncation of disordered terminal domains, conserved across plants and humans. Several novel proteoforms have been shown recombinantly to have distinct function from the full-length protein, but have not previously been shown to exist endogenously in cells. We additionally demonstrated that the C-terminus of the *Chlamydomonas* ortholog of Fibrocystin/PKHD1 localizes to the nucleus, mirroring the nuclear translocation of the proteolyzed Fibrocystin C-terminus in animals from the full-length membrane-localized protein. High-throughput, multispecies data from bottom-up mass spectrometry experiments thus allows the discovery and validation of conserved, abundant, stable, short proteoforms.

# Methods

## Co-fractionation/mass spectrometry

Human HEK293T cell (ATCC CRL3216, sex: female) size-exclusion CF/MS experiments were originally published in (Mallam et al, 2019), hemolysate CF/MS experiments were originally published in (Sae-Lee et al, 2022), and *Arabidopsis thaliana* and *Chlamydomonas reinhardtii* CF/MS experiments were originally published in (McWhite et al, 2020), and the experiments are described in full in those publications. Previously published CF/MS experiments (PXD013321, PXD013264, PXD013369 (McWhite et al, 2020), PXD015406 (Mallam et al, 2019), and PXD030050 (Sae-Lee et al, 2022) were retrieved from the PRIDE database (Perez-Riverol et al, 2019). Two additional human HEK293T cell (ATCC CRL3216, sex: female) ion exchange CF/MS experiments were collected as described in (Mallam et al, 2019), performing ion exchange chromatography as in (McWhite et al, 2020). These experiments have been deposited in the Massive/ProteomeXchange protein repository and are available under accession PXD036233. Table EV1 contains Massive/ProteomeXchange accessions for all experiments.

## Protein and peptide identification

Peptides were identified at a false discovery rate of 1% from each fraction using spectral matching algorithms MSGF+ (Kim and Pevzner, 2014), X!Tandem (Craig and Beavis, 2004), and Comet-2013020 (Eng et al, 2013), each run with 10ppm precursor tolerance, and allowing for fixed cysteine carbamidomethylation (+57.021464) and optional methionine oxidation (+15.9949), integrating peptide evidence across the search engines using MSBlender (Kwon et al, 2011). Spectra were compared to the UniProt (UniProt Consortium, 2019) human-reviewed reference proteome (20,191 proteins) along with 248 common contaminants,

considering up to two missed tryptic cleavages. We considered only peptides mapping uniquely to single proteins in a species (determined using the scripts trypsin.py and define_grouping.py in github.com/marcottelab/MS_grouped_lookup), and proteins with at least 10 unique peptides observed across all experiments.

## Identification of short proteoforms

A peptide profile matrix for each protein in each experiment was created, with columns corresponding to fractions, and rows for each identified peptide. Values were peptide spectral matches, scaled from 0 to 1 across each peptide row. Initially, we generated images of each protein's peptide elutions, and hand-selected images that showed irregular elution of N- and C-terminal peptides.

Proteins elute from columns with approximately Normal/Gaussian distribution across biochemical fractions. If a protein has multiple elution points due to either protein interactions or distinct proteoforms, the protein will elute approximately with a Multivariate Gaussian distribution. To identify multiple potential individual protein elution points, we fit a Multivariate Gaussian to each protein's elution for each experiment (Fig. 2A). This was done only for proteins with at least nine unique peptides observed in an experiment. We modified the AdaptGauss R package (Ultsch et al, 2015) to remove the Shiny application functionality and fit up to five Gaussian distributions, then applied it to each protein's peptide profile to identify multiple elution peaks, if present, basing the fit of the Multivariate Gaussian on minimizing root-mean-square deviation with no penalty for additional fitted peaks. We then separated individual elution peaks using intersections between adjacent Gaussian fits of the corresponding protein elution profile. This is a general method for segmenting multiple distinctly eluting protein species from a fractionation experiment.

To identify protein elution peaks with a skew towards either the N- or C-terminus, we calculated a terminal bias score (Fig. 2A) for each elution peak composed of at least seven unique peptides. For the peptides of each protein elution peak, a terminal bias score was calculated as described above to capture potential protein elution peaks composed of only peptides from either the N- or C-terminus of the protein. Briefly, this score captures the absence of observable peptides from either end of the protein. As a manual check, proteins with high max terminal bias scores were inspected with a custom R Shiny (Chang et al, 2022) application, and protein elution peaks manually assigned to either full-length, N-terminal, C-terminal, or internal proteoform categories.

For the case of human proteins, we compared the resulting proteoforms' sequence coverage to mRNA isoforms annotated in the Uniprot human reference proteome. We considered our proteoforms to be potentially derived from an mRNA isoform with a highly permissive criterion, specifically if the observed proteoform overlapped at least 50% the expected length of the expected mRNA isoform protein product.

## Plasmid construction, *Chlamydomonas* transformation, and microscopy

pYY025 (P_{PSAD}:PKHD1_C_Ter-CrNeonGreen-3FLAG:APHVIII) was constructed by isothermal recombination reaction (Gibson et al, 2009) of two fragments: (a) A DNA sequence encoding *Chlamydomonas* fibrocystin C-terminus (amino-acid positions 4415–4952), amplified from genomic DNA of wild-type strain CC-124 using the following primers: F: 5'- ATTTGCAGGAGATTCGAGGTTATGCGCGGCCGCCGCCAAGCGCCGC-3', R: 5'-CCTCGCCCTTGGACACCATGTTGTTGGCGTGGCGCTGGCGCATGCTGT-3' (1660 bps) and KOD Xtreme Hot Start DNA Polymerase (Toyobo, Cat No. 71975). (b) A bicistronic-mRNA-based expression vector pMO666 (P_{PSAD}:CrNeonGreen-3FLAG:APHVIII) digested with HpaI (Onishi and Pringle, 2016).

pYY025 was linearized using ScaI and transformed into wild-type strain CC-124 by electroporation as previously described (Onishi and Pringle, 2016), except that the concentration of paromomycin (RPI, Cat No. P11000) used for selection was reduced to 5 μg/ml.

Cells of positive transformants were cultured in 5 ml liquid Tris-acetate-phosphate (TAP) medium in a 100 ml glass jar at room temperature under constant illumination at 50–70 μmol photons m$^{-2}$ s$^{-1}$ for 2 days. Cells were mounted on glass slides with liquid TAP medium.

Images were taken with a Leica Thunder Cell Culture inverted microscope equipped with an HC PL APO 63X/1.40 N.A. oil-immersion objective lens. Signals were captured using the following excitation and emission wavelengths: 510 nm and 535/15 nm for mNeonGreen; 640 nm and 705/72 nm for chlorophyll autofluorescence. Fluorescence images were captured with 0.41 μm Z-spacing covering the entire cell volume; resulting images were processed through deconvolution and maximum-projected.

## Data availability

All code and data analysis are available from https://github.com/marcottelab/peptide_elutions/. HEK293T ion exchange CF/MS experiments have been deposited in the Massive/ProteomeXchange protein repository under accession PXD036233. Table EV1 contains Massive/ProteomeXchange accessions for all experiments.

The source data of this paper are collected in the following database record: biostudies:S-SCDT-10_1038-S44320-024-00048-3.

## Peer review information

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

## Acknowledgements

CDM acknowledges support from the Lewis-Sigler Institute for Integrative Genomics and NIH (F31 GM12384856). EMM acknowledges support from the Welch Foundation (F-1515), Army Research Office (W911NF-12-1-0390), and NIH (R35 GM122480, R01 HD085901). MO acknowledges partial support from NSF (MCB-1818383). The authors thank the Texas Advanced Computing Center at The University of Texas at Austin for providing high-performance computing resources that have contributed to the research results reported in this paper.

## Author contributions

**Claire D McWhite**: Conceptualization; Data curation; Software; Formal analysis; Investigation; Visualization; Methodology; Writing—original draft; Writing—review and editing. **Wisath Sae-Lee**: Resources; Investigation. **Yaning Yuan**: Validation. **Anna L Mallam**: Resources. **Nicolas A Gort-Freitas**: Formal analysis; Methodology. **Silvia Ramundo**: Validation; Methodology. **Masayuki Onishi**: Validation; Methodology; Writing—review and editing. **Edward M Marcotte**: Conceptualization; Resources; Supervision; Methodology; Writing—original draft; Project administration; Writing—review and editing.

Source data underlying figure panels in this paper may have individual authorship assigned. Where available, figure panel/source data authorship is listed in the following database record: biostudies:S-SCDT-10_1038-S44320-024-00048-3.

## Disclosure and competing interests statement

The authors declare no competing interests.

# Expanded View Figure

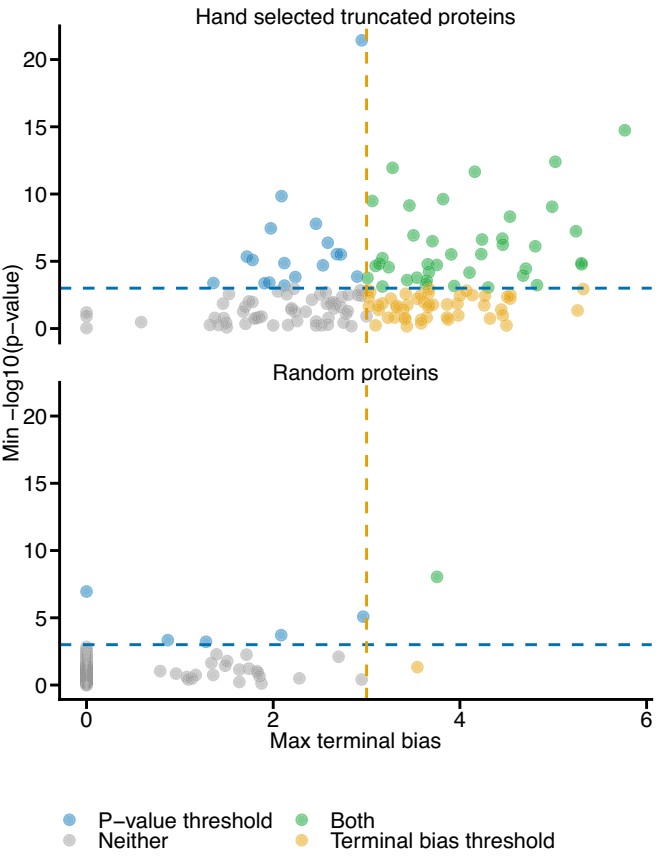

**Figure EV1.   Both maximum terminal bias score and minimum p-values prioritize proteins with apparent short proteoforms relative to randomly selected proteins.**

Proteins in green are prioritized by both methods, proteins in blue are prioritized by the *P* value threshold only, proteins in yellow are prioritized by the terminal bias threshold only, and those in gray by neither method.

