## [Peer Review File · Molecular Systems Biology]

Alternative proteoforms and proteoform-dependent assemblies in humans and plants

Claire McWhite, Wisath Sae-Lee, Yaning Yuan, Anna Mallam, Nicolas Gort-Freitas, Silvia Ramundo, Masayuki Onishi, and Edward Marcotte

Corresponding author(s): Claire McWhite (cmcwhite@princeton.edu)

Review Timeline:

Submission Date:	1st Feb 23
Editorial Decision:	14th Mar 23
Revision Received:	17th Jul 23
Editorial Decision:	22nd Aug 23
Revision Received:	4th Jun 24
Accepted:	6th Jun 24

Editor: Jingyi Hou and Maria Polychronidou

Transaction Report:

14th Mar 2023

Manuscript Number: MSB-2023-11575

Title: Alternative proteoforms and proteoform-dependent assemblies in humans and plants

Author: Claire McWhite

Wisath Sae-Lee

Yaning Yuan

Anna Mallam

Nicolas Gort-Freitas

Silvia Ramundo

Masayuki Onishi

Edward Marcotte

Dear Dr McWhite,

Thank you for submitting your work to Molecular Systems Biology. We have now heard back from two of the three reviewers who agreed to evaluate your manuscript. Unfortunately, after a series of reminders, we did not obtain a report from Reviewer #3. In the interest of time, and since the recommendations of the two reviewers are quite similar, I prefer to make a decision now rather than further delay the process. If we receive comments from Reviewer #3, we will forward them to you so that you can address any further issues raised. As you will see from the reports below, the reviewers acknowledge the interest of the study. They raise, however, a series of concerns, which we would ask you to address in a major revision.

The reviewers' recommendations are relatively straightforward, so there is no need to reiterate the points listed below. All the issues raised by the reviewers need to be satisfactorily addressed. As you may already know, our editorial policy allows in principle a single round of major revision, and it is therefore essential to provide responses to the reviewers' comments that are as complete as possible. Please feel free to contact me in case you would like to discuss in further detail any of the issues raised by the reviewers.

On a more editorial level, we would ask you to address the following issues:

- Please provide a .docx formatted version of the manuscript text (including legends for main figures, EV figures and tables). Please make sure that the changes are highlighted to be clearly visible.
- Please provide individual production quality figure files as .eps, .tif, .jpg (one file per figure).
- Please provide a .docx formatted letter INCLUDING the reviewers' reports and your detailed point-by-point responses to their comments. As part of the EMBO Press transparent editorial process, the point-by-point response is part of the Review Process File (RPF), which will be published alongside your paper.
- Please note that all corresponding authors are required to supply an ORCID ID for their name upon submission of a revised manuscript.
- We replaced Supplementary Information with Expanded View (EV) Figures and Tables that are collapsible/expandable online (see examples in <http://msb.embopress.org/content/11/6/812>). A maximum of 5 EV Figures can be typeset. EV Figures should be cited as 'Figure EV1, Figure EV2' etc... in the text and their respective legends should be included in the main text after the legends of regular figures.

Additional Tables/Datasets should be labeled and referred to as Table EV1, Dataset EV1, etc. Legends have to be provided in a separate tab in case of .xls files. Alternatively, the legend can be supplied as a separate text file (README) and zipped together with the Table/Dataset file.

For the figures and tables that you do NOT wish to display as Expanded View figures, they should be bundled together with their legends in a single PDF file called *Appendix*, which should start with a short Table of Content. Each legend should be below the corresponding Figure/Table in the Appendix. Appendix figures and tables should be referred to in the main text as: "Appendix Figure S1, Appendix Figure S2, Appendix Table S1" etc. See detailed instructions regarding expanded view here: <https://www.embopress.org/page/journal/17444292/authorguide#expandedview>.

- Before submitting your revision, primary datasets (and computer code, where appropriate) produced in this study need to be deposited in an appropriate public database (see <http://msb.embopress.org/authorguide> - dataavailability <https://www.embopress.org/page/journal/17444292/authorguide#dataavailability>).

The accession numbers and database should be listed in a formal "Data Availability" section (placed after Materials & Method) that follows the model below (see also <https://www.embopress.org/page/journal/17444292/authorguide#dataavailability>). Please note that the Data Availability Section is restricted to new primary data that are part of this study.

Data availability

-At EMBO Press we ask authors to provide source data for the main figures. Our source data coordinator will contact you to discuss which figure panels we would need source data for and will also provide you with helpful tips on how to upload and organize the files.

- Our journal encourages inclusion of *data citations in the reference list* to directly cite datasets that were re-used and obtained from public databases. Data citations in the article text are distinct from normal bibliographical citations and should directly link to the database records from which the data can be accessed. In the main text, data citations are formatted as follows: "Data ref: Smith et al, 2001". In the Reference list, data citations must be labeled with "[DATASET]". A data reference must provide the database name, accession number/identifiers and a resolvable link to the landing page from which the data can be accessed at the end of the reference. Further instructions are available at .

- We updated our journal's competing interests policy in January 2022 and request authors to consider both actual and perceived competing interests. Please review the policy <https://www.embopress.org/competing-interests> and update your competing interests if necessary.

Please use the heading "Disclosure statement and competing interests".

- All Materials and Methods need to be described in the main text. We would encourage you to use 'Structured Methods', our new Materials and Methods format. According to this format, the Material and Methods section should include a Reagents and Tools Table (listing key reagents, experimental models, software and relevant equipment and including their sources and relevant identifiers) followed by a Methods and Protocols section in which we encourage the authors to describe their methods using a step-by-step protocol format with bullet points, to facilitate the adoption of the methodologies across labs. More information on how to adhere to this format as well as downloadable templates (.doc or .xls) for the Reagents and Tools Table can be found in our author guidelines: < <https://www.embopress.org/page/journal/17444292/authorguide#researcharticleguide>>. An example of a Method paper with Structured Methods can be found here: .

-Regarding data quantification:

Please ensure to specify the name of the statistical test used to generate error bars and P values, the number (n) of independent experiments (please specify technical or biological replicates) underlying each data point and the test used to calculate p-values in each figure legend. Discussion of statistical methodology can be reported in the materials and methods section, but figure legends should contain a basic description of n, P and the test applied.

Graphs must include a description of the bars and the error bars (s.d., s.e.m.).

- Please provide a "standfirst text" summarizing the study in one or two sentences (approximately 250 characters, including space), three to four "bullet points" highlighting the main findings and a "synopsis image" (550px width and 400-600 px height, PNG format) to highlight the paper on our homepage.

Here are a couple of examples:

<https://www.embopress.org/doi/10.15252/msb.20199356>

<https://www.embopress.org/doi/10.15252/msb.20209475>

<https://www.embopress.org/doi/10.15252/msb.209495>

When you resubmit your manuscript, please download our CHECKLIST (<https://www.embopress.org/pb-assets/embosite/EMBO%20Press%20Author%20Checklist-1642513524327.xlsx>) and include the completed form in your submission.

Please note that the Author Checklist will be published alongside the paper as part of the transparent process (<https://www.embopress.org/page/journal/17444292/authorguide#transparentprocess>).

If you feel you can satisfactorily deal with these points and those listed by the referees, you may wish to submit a revised version of your manuscript. Please attach a covering letter giving details of the way in which you have handled each of the points raised

by the referees. A revised manuscript will be once again subject to review and you probably understand that we can give you no guarantee at this stage that the eventual outcome will be favorable.

I look forward to receiving your revised manuscript soon.

Sincerely,
Jingyi

Jingyi Hou, PhD
Scientific Editor
Molecular Systems Biology

We realize that it is difficult to revise to a specific deadline. In the interest of protecting the conceptual advance provided by the work, we recommend a revision within 3 months (12th Jun 2023). Please discuss the revision progress ahead of this time with the editor if you require more time to complete the revisions. Use the link below to submit your revision:

IMPORTANT: When you send your revision, we will require the following items:

1. the manuscript text in LaTeX, RTF or MS Word format
2. a letter with a detailed description of the changes made in response to the referees. Please specify clearly the exact places in the text (pages and paragraphs) where each change has been made in response to each specific comment given
3. three to four 'bullet points' highlighting the main findings of your study
4. a short 'blurb' text summarizing in two sentences the study (max. 250 characters)
5. a 'thumbnail image' (550px width and max 400px height, Illustrator, PowerPoint or jpeg format), which can be used as 'visual title' for the synopsis section of your paper.
6. Please include an author contributions statement after the Acknowledgements section (see <https://www.embopress.org/page/journal/17444292/authorguide>)
7. Please complete the CHECKLIST available at (<https://bit.ly/EMBOPressAuthorChecklist>). Please note that the Author Checklist will be published alongside the paper as part of the transparent process (<https://www.embopress.org/page/journal/17444292/authorguide#transparentprocess>).
8. When assembling figures, please refer to our figure preparation guideline in order to ensure proper formatting and readability in print as well as on screen:

See also figure legend guidelines: <https://www.embopress.org/page/journal/17444292/authorguide#figureformat>

9. Please note that corresponding authors are required to supply an ORCID ID for their name upon submission of a revised manuscript (EMBO Press signed a joint statement to encourage ORCID adoption).

(<https://www.embopress.org/page/journal/17444292/authorguide#editorialprocess>)

Currently, our records indicate that the ORCID for your account is 0000-0001-7346-3047.

Link Not Available

The system will prompt you to fill in your funding and payment information. This will allow Wiley to send you a quote for the article processing charge (APC) in case of acceptance. This quote takes into account any reduction or fee waivers that you may be eligible for. Authors do not need to pay any fees before their manuscript is accepted and transferred to the publisher.

EMBO Press participates in many Publish and Read agreements that allow authors to publish Open Access with reduced/no publication charges. Check your eligibility: <https://authorservices.wiley.com/author-resources/Journal-Authors/open-access/affiliation-policies-payments/index.html>

*** PLEASE NOTE *** As part of the EMBO Press transparent editorial process initiative (see our Editorial at

<https://dx.doi.org/10.1038/msb.2010.72>), Molecular Systems Biology publishes online a Review Process File with each accepted manuscripts. This file will be published in conjunction with your paper and will include the anonymous referee reports, your point-by-point response and all pertinent correspondence relating to the manuscript. If you do NOT want this File to be published, please inform the editorial office at msb@embo.org within 14 days upon receipt of the present letter.

Reviewer #1:

In "Alternative proteoforms and proteoform-dependent assemblies in humans and plants", Claire McWhite and co-authors describe a method for truncated proteoform identification that depends solely on bottom-up peptide analysis of fractionated lysate. This report describes a Gaussian mixture model that, when applied to data, can reveal a "terminal bias" suggesting the incidence of a truncated proteoform. Their methods were applied successfully to four different biological systems and a number of intriguing results were described. The presentation was clear, concise, thought-provoking, and delightful to read. I recommend that the editors accept this manuscript with major revisions, which I will describe below. This request for major revisions should not be an indicator of my lack of belief in the results or similarly a lack of interest on my part. I believe the results as presented and I find the paper to be a wonderful contribution to the field of proteomics.

Major revision:

1. Really, there is only one. That is that you didn't validate any of the findings with traditional top-down proteomics. I acknowledge that is hard. The sensitivity of top-down is poor and limited to low molecular weight proteoforms. Nonetheless, with isolated bands, this should be possible for at least a handful of your targets. I rarely request additional experimental evidence in my manuscript reviews because of the tremendous burden it places on students and labs. However, in this instance, I think my request is warranted. Validation of your results by intact proteoform analysis would really strengthen the conclusions stated.

Minor revisions:

1. Experiment description accompanying Figure 1 in the text and in the caption are not clear. I understand this is a co-fractionation experiment. I couldn't tell if it was size exclusion or ion exchange. Moreover, if the bands are from different complexes, they are not necessarily, truncated proteoforms. I am sure this is a misunderstanding on my part. But, more clear text would aid in understanding.
2. The GitHub link provided in the manuscript is broken.
3. All data should be submitted to PRIDE or MASSIVE.

Reviewer #2:

In their manuscript "Alternative proteoforms and proteoform-dependent assemblies in humans and plants", McWhite and colleagues describe an explorative study focused on the identification of proteoforms using new and previously published cofraction mass spectrometry (CF-MS) experimental designs. They first develop a tailored analysis approach for their problem and then apply it to a selection of suitable datasets. For each dataset, the resulting "hits" (0-2% of total proteins) are then assessed in more detail, grouped by putative biological mechanisms underlying proteoform generation, and then described in an anecdotal fashion.

I believe their research topic is very exciting and that such studies could lead further investigations in this area. However, there are substantial technical challenges, which the authors also acknowledge, that need to be overcome to increase the usefulness of these efforts. Further, the manuscript lacks systematic interpretation or the description on how the examples were selected.

While I think their idea and application will be of interest to the community, more technical improvements and guidance on the systematic interpretation of such results are required.

Major points:

1. Terminal bias score

The authors develop a heuristic terminal bias score that allows to identify alternative proteoforms. They demonstrate that the score is selective, but extremely biased towards abundant proteins with many observed peptides. Cumulatively, it allows to detect 0-2% alternative proteoforms per CF-MS dataset. The authors acknowledge these limitations, but it remained unclear to me whether these limitations are due to CF-MS or the chosen analysis approach. The authors should investigate or at least discuss the following issues:

a) Statistical significance

The terminal bias score provides a raw metric to differentiate between untruncated vs truncated proteoforms. Although it is selective, no measure for significance is provided, which is key to interpret any high-throughput results systematically. The authors should thus investigate whether confidence metrics could be implemented or they at least should discuss alternative options to control error-rate in high-throughput screens.

b) Comparison with other approaches used in over-representation analysis

To put their method into context, the authors should compare the performance against established methods from over-representation analysis (Fisher's exact test, GSEA, etc.). Essentially, they could test whether C- or N-terminal peptides are overrepresented in the set of observed peptides. In addition to the test statistic, these tests also provide a measure of confidence that could be useful. If the tests don't work, this finding would also support the use of their terminal bias score.

For GSEA, a potential extension could include accounting for predicted observability (e.g. from APEX, AlphaPeptDeep), which might help to increase recovery of alternative proteoforms with fewer measured peptides.

2. Orthogonal, MW-based scoring

Currently, the analysis strategy does not seem to include or consider the apparent molecular weight of the measured fractions. Although many alternative proteoforms might be measured as part of complexes, and thus not be available in monomeric conformation, it would still be interesting to have a sense of accuracy for those that fall within the expected window of their monomeric masses. Specifically, the authors should compute and provide the fraction of alternative proteoforms, which can be found at their expected molecular weight in relation to those outside this window.

3. Guidance on systematic interpretation

The majority of the results section of the manuscript then focuses on the discussion of specific results, grouped according to the putative biological mechanisms underlying proteoform generation. However, it is not clear to me how the criteria to select the examples were selected. Was this done by simply going through the top candidate list and grouping potential hits according to arbitrary rules? The authors should clarify this strategy, so future studies can follow a similar rationale.

a) In each section, briefly describe the analysis criteria to include examples in the group. How many candidates above the terminal bias score cutoff fall into each group? Is there any experimental or analysis bias that should be considered?

b) Either at the beginning or end of the results section, the authors should provide a systematic listing of the results. How many candidates were detected and to which subgroups (including overlap) do they belong? It's clear that the method has extreme biases towards certain proteins only, but nevertheless, it would be interesting to understand the statistics of the discoveries a bit more in a cumulative sense.

Minor points:

4. Methods section

In a similar vein, the methods section only captures the computation of the terminal bias score, but no criteria for the identification of the groups. While this would allow to reproduce their main scores, I doubt that it would be possible to reproduce their grouping or the selection of specific alternative proteoforms described in the paper.

5. Source code accessibility

The source code linked by the authors (https://github.com/marcottelab/peptide_elutions/) was not available for review. It's thus not possible to judge whether their algorithms can easily be adapted for future studies. The authors should either make the code publicly available or provide a copy via the supplemental material.

Responses to referees

Our responses follow in line in blue text

Reviewer #1:

In "Alternative proteoforms and proteoform-dependent assemblies in humans and plants", Claire McWhite and co-authors describe a method for truncated proteoform identification that depends solely on bottom-up peptide analysis of fractionated lysate. This report describes a Gaussian mixture model that, when applied to data, can reveal a "terminal bias" suggesting the incidence of a truncated proteoform. Their methods were applied successfully to four different biological systems and a number of intriguing results were described. The presentation was clear, concise, thought-provoking, and delightful to read. I recommend that the editors accept this manuscript with major revisions, which I will describe below. This request for major revisions should not be an indicator of my lack of belief in the results or similarly a lack of interest on my part. I believe the results as presented and I find the paper to be a wonderful contribution to the field of proteomics."

We thank the reviewer for their positive and encouraging remarks on our manuscript's presentation and contribution to proteomics.

Major revision:

1. Really, there is only one. That is that you didn't validate any of the findings with traditional top-down proteomics. I acknowledge that is hard. The sensitivity of top-down is poor and limited to low molecular weight proteoforms. Nonetheless, with isolated bands, this should be possible for at least a handful of your targets. I rarely request additional experimental evidence in my manuscript reviews because of the tremendous burden it places on students and labs. However, in this instance, I think my request is warranted. Validation of your results by intact proteoform analysis would really strengthen the conclusions stated.

Demonstrating that truncated proteoforms exist *in vivo* can be intrinsically hard because of, for example, a lack of affinity reagents that distinguish the potential proteoforms and the potential instability of certain proteins outside of the cellular environment, among many other factors. Validation specifically by top-down proteomics is not just hard, but an *exceptional* challenge. The referee indicates that we have isolated bands—however, this is perhaps a misreading of our figures and most definitely not the case here. We would like to clarify that what is represented is not bands of isolated proteins, but a heatmap of fractions where a peptide from a particular protein was detected. Importantly, we do not purify *any* of the short proteoforms prior to analysis. Instead, the power of our approach is the ability to look directly at complex samples, and the samples we analyze are full complexity protein extracts from cell lysates, subjected to a single native chromatographic separation with the fractions subsequently analyzed by mass spectrometry. Each mass spec sample is ~1/40 the complexity of full cell lysate, far beyond the capacity of conventional top-down proteomics to handle without extraordinary efforts.

Thus, top-down proteomics is simply not an option for most of our samples due to the high (and highly variable) complexity of the samples, with the single exception of red blood cells, for which an independent top-down dataset already exists (see below). Consequently, instead of techniques involving protein purification, we prefer to use the hierarchy of evidence in Figure 8C, and to address the referee's comments, we have now added additional evidence in support of our findings.

Note that we already present a high level of overlap between our candidate proteoforms and multiple examples of known proteoforms from the literature. However, we did have an opportunity to compare our results to top-down data specifically for red blood cells by taking advantage of data from the Blood Proteoform Atlas (PMID: 35084980), a large-scale top-down analysis of different blood cell types. The BPA contains top-down measurements of erythrocytes, while our data specifically documents hemolysate (the soluble portion of lysed erythrocytes), so we examined those proteins present in both datasets.

Limiting the BPA proteoforms to only those from proteins we detect in hemolysate, the BPA finds 279 length-variant proteoforms from 57 proteins; 75 of the proteoforms derived from a single protein, the beta subunit of hemoglobin. Our analysis, on hemolysate from which hemoglobin was first depleted using HemogloBind, revealed 33 proteoforms from 29 proteins, 3 of which were shared between the two datasets. Among the three proteins with proteoforms shared between the two data sets, we find evidence in ours for previously observed and named portions of ANK1 ("55kDa regulatory domain", PMID: 3038887) and Band 4.1 ("22-24 kDa CTD domain", PMID: 11432737); neither were present in the top-down set. Thus, we can provide some degree of confidence that our candidate proteoforms are enriched for validated examples of stable protein fragments. Our conclusion is that the two proteomics techniques are complementary, each offering the opportunity to discover legitimate proteoforms, but each with limited coverage.

We now describe this analysis on p. 8 of the manuscript.

Minor revisions:

1. Experiment description accompanying Figure 1 in the text and in the caption are not clear. I understand this is a co-fractionation experiment. I couldn't tell if it was size exclusion or ion exchange. Moreover, if the bands are from different complexes, they are not necessarily, truncated proteoforms. I am sure this is a misunderstanding on my part. But, more clear text would aid in understanding.

We're sorry for the confusion. We have modified the introduction to first explain that we use peptide coverage to determine proteoforms, and only use separation to allow proteoforms with different peptide coverage to be distinguished from each other. Some of our data comes from size exclusion experiments and some comes from ion exchange. However, our figure only illustrates the concept of biochemical separation of a complex lysate into fractions so that the

peptide coverage of individual protein species can be examined, *i.e.* that fractionation allows proteoforms with different peptide coverage to be distinguished from one another.

We have now added text to the introduction as well as to the in-text description of Figure 1E, describing how we distinguish between a full length protein eluting at multiple points and a shortened version of a protein.

Specifically, a protein may elute at multiple points in the fractionation based on complex membership or multimerization, but if it has full peptide coverage, we do not call it a proteoform. We only consider it a truncation proteoform if it is missing peptide coverage from one or the other terminus and where such peptides are known to be detectable from other proteoforms.

2. The GitHub link provided in the manuscript is broken.

Fixed. We had neglected to change the repository from private to public.

3. All data should be submitted to PRIDE or MASSIVE.

Supplemental File 1 now contains Massive/ProteomeXchange accessions for all experiments.

Reviewer #2:

In their manuscript "Alternative proteoforms and proteoform-dependent assemblies in humans and plants", McWhite and colleagues describe an explorative study focused on the identification of proteoforms using new and previously published cofraction mass spectrometry (CF-MS) experimental designs. They first develop a tailored analysis approach for their problem and then apply it to a selection of suitable datasets. For each dataset, the resulting "hits" (0-2% of total proteins) are then assessed in more detail, grouped by putative biological mechanisms underlying proteoform generation, and then described in an anecdotal fashion.

I believe their research topic is very exciting and that such studies could lead further investigations in this area. However, there are substantial technical challenges, which the authors also acknowledge, that need to be overcome to increase the usefulness of these efforts. Further, the manuscript lacks systematic interpretation or the description on how the examples were selected.

We thank the reviewer for their encouraging assessment. We have now added additional text clarifying how examples were chosen in the paragraph immediately before the examples begin (section "Short proteoform boundaries occur preferentially outside of domains"). Briefly, we chose examples 1) based on correspondence to known proteoforms from the literature as "positive controls" to establish that our method recovers known proteoforms, 2) biological interest, and 3) proteoform-specific protein interactions.

“While I think their idea and application will be of interest to the community, more technical improvements and guidance on the systematic interpretation of such results are required.”

We think our approach is complementary in many ways to previous methods for proteoform discovery, and will be useful towards the goal of characterizing the full extent of protein diversity. To aid in systematic interpretation, we have now added a precision-recall curve to our Figure 2 results that demonstrates the predictive power of the terminal bias score in prioritizing proteins with candidate truncation proteoforms.

Major points:

1. Terminal bias score

The authors develop a heuristic terminal bias score that allows to identify alternative proteoforms. They demonstrate that the score is selective, but extremely biased towards abundant proteins with many observed peptides. Cumulatively, it allows to detect 0-2% alternative proteoforms per CF-MS dataset. The authors acknowledge these limitations, but it remained unclear to me whether these limitations are due to CF-MS or the chosen analysis approach. The authors should investigate or at least discuss the following issues:

We have now added a further discussion of the limitations of detecting alternate proteoforms using this technique to the conclusion of the paper. Our main limitation in detecting truncated proteins is in peptide coverage/protein abundance. As suggested by the reviewer in their comment below, deep learning methods that increase peptide recovery would increase the number of proteoforms that are detectable by this method. When peptide coverage is sufficient, truncated proteoforms can easily be distinguished by eye. While we use the heuristic to prioritize proteins for visual inspection, and in the future will work toward an improved scoring method that fully matches expert curation.

a) Statistical significance

The terminal bias score provides a raw metric to differentiate between untruncated vs truncated proteoforms. Although it is selective, no measure for significance is provided, which is key to interpret any high-throughput results systematically. The authors should thus investigate whether confidence metrics could be implemented or they at least should discuss alternative options to control error-rate in high-throughput screens.

We have now calculated a precision-recall curve of the terminal bias score in prioritizing proteins with candidate truncated proteoforms, and include this curve as a new Figure 2C. We compared the terminal bias score for 278 hand selected proteins with truncations to 1000 random proteins. Above a terminal bias score of 1, the true positive rate is 71.6% with a recall of 92.4%, above a score of 2, the true positive rate is 90.7% with recall of (77.3%); above a score of 4, the true positive rate is 100% with a recall of 17.3%.

New Figure 2C: Precision and recall above threshold terminal bias scores

b) Comparison with other approaches used in over-representation analysis

To put their method into context, the authors should compare the performance against established methods from over-representation analysis (Fisher's exact test, GSEA, etc.). Essentially, they could test whether C- or N-terminal peptides are overrepresented in the set of observed peptides. In addition to the test statistic, these tests also provide a measure of confidence that could be useful. If the tests don't work, this finding would also support the use of their terminal bias score.

For truncation variants, the unknown exact N- or C-terminal position of our candidate proteoforms introduces difficulties for confident peptide detection. Additionally, even canonical C-terminal peptides are difficult to detect in complex mixtures after trypsin digest, due to the lack of a charged Arg/Lys at the C-terminus of the final C-terminal peptide (unless naturally present at the C-terminus of the protein). However, this is rare, as only about 16% of human proteins naturally end in a positively charged amino acid.

To overcome this, other methods such as the COFRADIC terminomics approach to identify truncation variants first enrich for N- and C-terminal amino acids (PMID: 28315247). Alternatively, use of LysargiNase to digest proteins improves detection of C-terminal amino acids (PMID: 25419962). However, without enrichment, considering the rarity of C-terminal terminal amino acids, and also the unknown exact start/end of N- or C-terminal candidate truncated proteoforms, there is just not enough information available to us for an analysis of terminal peptide enrichment. We have however now added to the conclusion that our method does not require enrichment of terminal peptides.

For GSEA, a potential extension could include accounting for predicted observability (e.g. from APEX, AlphaPeptDeep), which might help to increase recovery of alternative proteoforms with fewer measured peptides.

We definitely agree that increased peptide recovery would boost the ability to recover alternative proteoforms, and have added references to the ability of these methods to improve peptide identifications to our concluding paragraph.

2. Orthogonal, MW-based scoring

Currently, the analysis strategy does not seem to include or consider the apparent molecular weight of the measured fractions. Although many alternative proteoforms might be measured as part of complexes, and thus not be available in monomeric conformation, it would still be interesting to have a sense of accuracy for those that fall within the expected window of their monomeric masses. Specifically, the authors should compute and provide the fraction of alternative proteoforms, which can be found at their expected molecular weight in relation to those outside this window.

While this suggestion seems like a clear orthogonal support for proteoforms, the relationship between mass and elution position in size exclusion chromatography is highly complex, and is influenced by other factors beyond size including hydrodynamic radius and electrostatic interactions (PMID: 9540208). Size calibration in native SEC is additionally complicated by interactions between proteins and differential behavior of globular vs. intrinsically disordered proteins (PMID: 23269364). Due to both this inexact dependence of size and the unknown multimeric state of both intact proteins and truncated proteoforms, native size exclusion chromatography is not directly useful for making exact statements on molecular weight of observed proteoforms.

However, we do find that truncated proteoforms typically elute at later positions than their full-length versions. In HEK size exclusion experiments where both a full length and short form of a protein are observed, the full length proteoform elutes first 73% of the time, as we now note on page 7. Interestingly, of short proteoforms that elute before the full length version, the majority are intrinsically disordered termini. While biophysical properties of intrinsically disordered proteins are complex, they do tend to elute faster than globular proteins due to their larger Stoke's radius (PMID: 23269364).

22nd Aug 2023

Manuscript Number: MSB-2023-11575R

Title: Alternative proteoforms and proteoform-dependent assemblies in humans and plants

Dear Dr McWhite,

Thank you for sending us your revised manuscript. We have now heard back from the two reviewers who were asked to evaluate your revised study. As you will see below, the reviewers think that the study has improved as a result of the performed revisions. However, reviewer #2 still lists a remaining issue which we would ask you to address in a revision, alongside some editorial issues listed below..

- Our data editors have noted some missing information in the figure legends, please see the attached .doc file. We have also made some minor formatting changes. Please make all requested text changes using the attached file and *keeping the "track changes" mode* so that we can easily access the edits made.

- Please include callouts for Fig 7C and 7D in the main text.

- Supplemental Files 1 and 2 should be named Table EV1 and Dataset EV1 respectively. Please include the description of each Table/Dataset in the xls file itself (as a separate sheet) and remove them from the manuscript text.

- Please format the References according to the Molecular Systems Biology reference style i.e. ordered alphabetically and listing the first 10 authors followed by et al. The DOIs of published papers should be removed from the Reference list.

Please resubmit your revised manuscript online, with a covering letter listing amendments and responses to each point raised by the referees. Please resubmit the paper ****within one month**** and ideally as soon as possible. If we do not receive the revised manuscript within this time period, the file might be closed and any subsequent resubmission would be treated as a new manuscript. Please use the Manuscript Number (above) in all correspondence.

Click on the link below to submit your revised paper.

Kind regards,

Maria

Maria Polychronidou, PhD
Senior Editor
Molecular Systems Biology

If you do choose to resubmit, please click on the link below to submit the revision online before 21st Sep 2023.

IMPORTANT: When you send your revision, we will require the following items:

1. the manuscript text in LaTeX, RTF or MS Word format
2. a letter with a detailed description of the changes made in response to the referees. Please specify clearly the exact places in the text (pages and paragraphs) where each change has been made in response to each specific comment given
3. three to four 'bullet points' highlighting the main findings of your study
4. a short 'blurb' text summarizing in two sentences the study (max. 250 characters)
5. a 'thumbnail image' (550px width and max 400px height, Illustrator, PowerPoint or jpeg format), which can be used as 'visual title' for the synopsis section of your paper.
6. Please include an author contributions statement after the Acknowledgements section (see <https://www.embopress.org/page/journal/17444292/authorguide#manuscriptpreparation>)

7. Please complete the CHECKLIST available at (<https://bit.ly/EMBOPressAuthorChecklist>).

Please note that the Author Checklist will be published alongside the paper as part of the transparent process (<https://www.embopress.org/page/journal/17444292/authorguide#transparentprocess>).

See also figure legend guidelines: <https://www.embopress.org/page/journal/17444292/authorguide#figureformat>

9. Please note that corresponding authors are required to supply an ORCID ID for their name upon submission of a revised manuscript (EMBO Press signed a joint statement to encourage ORCID adoption).

(<https://www.embopress.org/page/journal/17444292/authorguide#editorialprocess>)

Currently, our records indicate that the ORCID for your account is 0000-0001-7346-3047.

Link Not Available

The system will prompt you to fill in your funding and payment information. This will allow Wiley to send you a quote for the article processing charge (APC) in case of acceptance. This quote takes into account any reduction or fee waivers that you may be eligible for. Authors do not need to pay any fees before their manuscript is accepted and transferred to the publisher.

*** PLEASE NOTE *** As part of the EMBO Press transparent editorial process initiative (see our Editorial at <https://dx.doi.org/10.1038/msb.2010.72> , Molecular Systems Biology will publish online a Review Process File to accompany accepted manuscripts. When preparing your letter of response, please be aware that in the event of acceptance, your cover letter/point-by-point document will be included as part of this File, which will be available to the scientific community. More information about this initiative is available in our Instructions to Authors. If you have any questions about this initiative, please contact the editorial office (msb@embo.org).

Reviewer #1:

The revisions satisfactorily address my concerns

Reviewer #2:

In their revised manuscript, the authors have addressed several of the raised issues. Most importantly, they now provide precision-recall curves for their "terminal bias score".

However, they have not further investigated my suggestion to compare their "terminal bias score" against standard over-representation analysis (ORA) approaches. In my original comment, I did not intend to suggest that additional or better data would be required. Instead, I was curious how standard approaches used in statistics would perform compared to their heuristic score for this particular challenge. For example, Fisher's exact test could be simply applied to the example visualized in Fig. 2A comparing the binary peptide identification profiles of N-term vs Full length Protein A or C-term vs Full length Protein A. The benefit of this analysis is two-fold: First, if the approach at least partially works, it provides a second, readily-interpretable standard metric for data interpretation. Second, if the approach does not work, it strongly supports using (and in the future extending) their "terminal bias score".

Reviewer #2:

In their revised manuscript, the authors have addressed several of the raised issues. Most importantly, they now provide precision-recall curves for their "terminal bias score".

However, they have not further investigated my suggestion to compare their "terminal bias score" against standard over-representation analysis (ORA) approaches. In my original comment, I did not intend to suggest that additional or better data would be required. Instead, I was curious how standard approaches used in statistics would perform compared to their heuristic score for this particular challenge. For example, Fisher's exact test could be simply applied to the example visualized in Fig. 2A comparing the binary peptide identification profiles of N-term vs Full length Protein A or C-term vs Full length Protein A. The benefit of this analysis is two-fold: First, if the approach at least partially works, it provides a second, readily-interpretable standard metric for data interpretation. Second, if the approach does not work, it strongly supports using (and in the future extending) their "terminal bias score".

We thank the reviewer for the suggestion to compare our terminal bias score with a standard over-representation analysis approach. We have now conducted Fisher's exact tests on all observed proteins as suggested and have updated Figures 2A and 2B to include Fisher's exact test results, providing an additional method to prioritize proteins that may exhibit interesting candidate truncation patterns.

Upon analysis, we found that the p-values from Fisher's exact test and our terminal bias score tend to agree on many cases, yet each uniquely recovers interesting truncations that the other misses (as shown in New Figure EV1 accompanying main text Figure 2). While neither score alone can completely recover peptide profiles indicating potential truncations, the terminal bias score tends to better prioritize proteins with interesting truncation patterns (as depicted in Figure 2B). We hope to create a more effective metric for identifying proteins with potential truncation patterns that integrates the strengths of both the terminal bias score and ORA.

However, since the terminal bias score and Fisher's exact test each uniquely contribute to identifying candidate truncations, we recognize the importance of leveraging both metrics. To enhance the robustness of our analysis, we have now included Benjamini-Hochberg corrected p-values from Fisher's exact tests in our supplementary file of identified candidate truncated proteins (Dataset EV1). This provides a second measure of confidence for data interpretation and supports our ongoing efforts to develop an improved metric for scoring.

6th Jun 2024

Manuscript number: MSB-2023-11575RR

Title: Alternative proteoforms and proteoform-dependent assemblies in humans and plants

Thank you again for sending us your revised manuscript. We are now satisfied with the modifications made and I am pleased to inform you that your paper has been accepted for publication.

Yours sincerely,
Jingyi

Jingyi Hou, PhD
Scientific Editor
Molecular Systems Biology
